# Forming Scalable, Convergent GNN Layers that Minimize a Sampling-Based Energy

## Abstract

Among the many variants of graph neural network (GNN) architectures capable of modeling data with cross-instance relations, an important subclass involves layers designed such that the forward pass iteratively reduces a graph-regularized energy function of interest. In this way, node embeddings produced at the output layer dually serve as both predictive features for solving downstream tasks (e.g., node classification) and energy function minimizers that inherit transparent, exploitable inductive biases. However, scaling GNN architectures constructed in this way remains challenging, in part because the convergence of the forward pass may involve models with considerable depth. To tackle this limitation, we propose a sampling-based energy function and scalable GNN layers that iteratively reduce it, guided by convergence guarantees in certain settings. We also instantiate a full GNN architecture based on these designs, and the model achieves competitive accuracy and scalability when applied to the largest publicly-available node classification benchmark exceeding 1TB in size.

## 1 Introduction

Graph neural networks (GNNs) are a powerful class of deep learning models designed specifically for graph-structured data. Unlike conventional neural networks that primarily operate on independent samples, GNNs excel in capturing the complex cross-instance relations modeled by graphs (Hamilton et al., 2017; Kearnes et al., 2016; Kipf & Welling, 2017; Velickovic et al., 2018). Foundational to GNNs is the notion of message passing (Gilmer et al., 2017), whereby nodes iteratively gather information from neighbors to update their representations. In doing so, information can propagate across the graph in the form of node embeddings, reflecting both local patterns and global network effects, which are required by downstream tasks such as node classification.

Among many GNN architectures, one promising subclass is based on graph propagation layers derived to be in a one-to-one correspondence with the descent iterations of a graph-regularized energy function (Chen et al., 2022; Ahn et al., 2022; Yang et al., 2021; Ma et al., 2021; Gasteiger et al., 2019; Pan et al., 2020; Zhang et al., 2020; Zhu et al., 2021; Xue et al., 2023). For these models, the layers of the GNN forward pass computes increasingly-refined approximations to a minimizer of the aforementioned energy. Importantly, if such energy minimizers possess desirable properties or inductive biases, the corresponding GNN architecture naturally inherits them, unlike traditional GNN constructions that may be less transparent. More broadly, this association between the GNN forward pass and optimization can be exploited to introduce targeted architectural enhancements (e.g., robustly handling graphs with spurious edges and/or heterophily (Fu et al., 2023; Yang et al., 2021). Borrowing from (Chen & Eldar, 2021; Yang et al., 2022), we will henceforth refer to models of this genre as *unfolded* GNNs, given that the layers are derived from a so-called unfolded (in time/iteration) energy descent process.

Despite their merits w.r.t. transparency, unfolded GNNs face non-trivial scalability challenges. This is in part because they can be constructed with arbitrary depth (i.e., arbitrary descent iterations) while still avoiding undesirable oversmoothing effects (Oono & Suzuki, 2020; Li et al., 2018), and the computational cost and memory requirements of this flexibility are often prohibitively high. Even so, there exists limited prior work explicitly tackling the scalability of unfolded GNNs, or providing any complementary guarantees regarding convergence on large graphs. Hence most unfolded GNN models are presently evaluated on relatively small benchmarks.

To address these shortcomings, we propose a scalable unfolded GNN model that incorporates efficient subgraph sampling within the fabric of the requisite energy function. Our design of this model

is guided by the following three desiderata: (i) maintain the characteristic properties and extensible structure of full-graph unfolded GNN models, (ii) using a consistent core architecture, preserve competitive accuracy across datasets of varying size, including the very largest publicly-available benchmarks, and (iii) do not introduce undue computational or memory overheads beyond what is required to train the most common GNN alternatives. Our proposed model, which we will later demonstrate satisfies each of the above, is termed *MuseGNN* in reference to a GNN architecture produced by the *minimization* of an *unfolded sampling*-based *energy*. Building on background motivations presented in Section 2, our MuseGNN-related contributions herein are three-fold:

- In Sections 3 and 4, we expand a widely-used unfolded GNN framework to incorporate offline sampling into the architecture-inducing energy function design itself, as opposed to a post hoc application of sampling to existing GNN methods. The resulting model, which we term MuseGNN, allows us to retain the attractive attributes of unfolded GNNs, and an unbiased estimator of the original full-graph energy, with the scalability of canonical GNN architectures.

- We analytically demonstrate in Section 5 that MuseGNN possesses desirable convergence properties regarding both upper-level (traditional model training) and lower-level (energy function descent) optimization processes. In doing so we increase our confidence in reliable performance when moving to new problem domains that may deviate from known benchmarks.

- Finally, in Section 6 we provide complementary empirical support that MuseGNN performance is stable in practice, preserving competitive accuracy and scalability across task size. En route, we achieve SOTA performance w.r.t. homogeneous graph models applied to the largest, publicly-available node classification datasets from OGB and IGB exceeding 1TB.

## 2 BACKGROUND AND MOTIVATION

This section provides background details regarding existing unfolded GNN architectures. Later, we discuss why GNNs formed in this way are relevant, followed by their scalability challenges.

### 2.1 GNN ARCHITECTURES FROM UNFOLDED OPTIMIZATION

**Notation.** Let $\mathcal{G} = \{\mathcal{V}, \mathcal{E}\}$ denote a graph with $n = |\mathcal{V}|$ nodes and edge set $\mathcal{E}$. We define $D$ and $A$ as the degree and adjacency matrices of $\mathcal{G}$ such that the corresponding graph Laplacian is $L = D - A$. Furthermore, associated with each node is both a $d$-dimensional feature vector, and a $d'$-dimensional label vector, the respective collections of which are given by $X \in \mathbb{R}^{n \times d}$ and $T \in \mathbb{R}^{n \times d'}$.

**GNN Basics.** Given a graph defined as above, canonical GNN architectures are designed to produce a sequence of node-wise embeddings $\{Y^{(k)}\}_{k=1}^K$ that are increasingly refined across $K$ model layers according to the rule $Y^{(k)} = h(Y^{(k-1)}; W, A, X) \in \mathbb{R}^{n \times d}$. Here $h$ represents a function that updates $Y^{(k-1)}$ based on trainable model weights $W$ as well as $A$ (graph structure) and optionally $X$ (input node features). To facilitate downstream tasks such as node classification, $W$ may be trained, along with additional parameters $\theta$ of any application-specific output layer $g : \mathbb{R}^d \to \mathbb{R}^{d'}$, to minimize a loss of the form

$$\mathcal{L}(W, \theta) = \sum_{i=1}^{n'} \mathcal{D}\bigg( g\left[Y^{(K)}(W)_i; \theta\right], T_i \bigg). \tag{1}$$

In this expression, $Y_i^{(K)} \equiv Y^{(K)}(W)_i$ reflects the explicit dependency of GNN embeddings on $W$ and the subscript $i$ denotes the $i$-th row of a matrix. Additionally, $n'$ refers to the number of nodes in $\mathcal{G}$ available for training, while $\mathcal{D}$ is a discriminator function such as cross-entropy. We will sometimes refer to the training loss (1) as an *upper-level* energy function to differentiate its role from the lower-level energy defined next.

**Moving to Unfolded GNNs.** An *unfolded* GNN architecture ensues when the functional form of $h$ is explicitly chosen to align with the update rules that minimize a second, *lower-level* energy function denoted as $\ell(Y)$. In this way, it follows that $\ell\left(Y^{(k)}\right) = \ell\left(h\left[Y^{(k-1)}; W, A, X\right]\right) \leq \ell\left(Y^{(k-1)}\right)$, This restriction on $h$ is imposed to introduce desirable inductive biases on the resulting GNN layers that stem from properties of the energy $\ell$ and its corresponding minimizers (see Section 2.2 ).

While there are many possible domain-specific choices for $\ell(Y)$ in the growing literature on unfolded GNNs (Chen et al., 2022; Ahn et al., 2022; Ma et al., 2021; Gasteiger et al., 2019; Pan et al., 2020; Yang et al., 2021; Zhang et al., 2020; Zhu et al., 2021; Xue et al., 2023), we will focus our attention on a particular form that encompasses many existing works as special cases, and can be easily generalized to cover many others. Originally inspired by Zhou et al. (2003) and generalized by Yang et al. (2021) to account for node-wise nonlinearities, we consider

$$\ell(Y) := \|Y - f(X;W)\|_F^2 + \lambda \operatorname{tr}(Y^\top L Y) + \sum_{i=1}^{n} \zeta(Y_i), \qquad (2)$$

where $\lambda > 0$ is a trade-off parameter, $f$ represents a trainable base model parameterized by $W$ (e.g., a linear layer or MLP), and the function $\zeta$ denotes a (possibly non-smooth) penalty on individual node embeddings. The first term in (2) favors embeddings that resemble the input features as processed by the base model, while the second encourages smoothness across graph edges. And finally, the last term is included to enforce additional node-wise constraints (e.g., non-negativity).

We now examine the form of $h$ that can be induced when we optimize (2). Although $\zeta$ may be non-smooth and incompatible with vanilla gradient descent, we can nonetheless apply proximal gradient descent in such circumstances (Combettes & Pesquet, 2011), leading to the descent update

$$Y^{(k)} = h\left(Y^{(k-1)}; W, A, X\right) = \operatorname{prox}_\zeta\left(Y^{(k-1)} - \alpha\left[(I + \lambda L)Y^{(k-1)} - f(X;W)\right]\right), \qquad (3)$$

where $\alpha$ controls the learning rate and $\operatorname{prox}_\zeta(u) := \arg\min_y \frac{1}{2}\|u - y\|_2^2 + \zeta(y)$ denotes the proximal operator of $\zeta$. Analogous to more traditional GNN architectures, (3) contains a $\zeta$-dependent nonlinear activation applied to the output of an affine, graph-dependent filter. With respect to the former, if $\zeta$ is chosen as an indicator function that assigns an infinite penalty to any embedding less than zero, then $\operatorname{prox}_\zeta$ reduces to standard ReLU activations; we will adopt this choice for MuseGNN.

## 2.2 Why Unfolded GNNs?

There are a variety of reasons why it can be advantageous to construct $h$ using unfolded optimization steps as in (3) or related. Of particular note here, unfolded GNN node embeddings inherit exploitable characteristics of the lower-level energy, especially if $K$ is sufficiently large such that $Y^{(K)}$ approximates a minimum of $\ell(Y)$. For example, it is well-known that an oversmoothing effect can sometimes cause GNN layers to produce node embeddings that converge towards a non-informative constant (Oono & Suzuki, 2020; Li et al., 2018). This can be understood through the lens of minimizing the the second term of (2) in isolation (Cai & Wang, 2020). The latter can be driven to zero whenever $Y_i^{(K)} = Y_j^{(K)}$ for all $i, j \in \mathcal{V}$, since $\operatorname{tr}[(Y^{(K)})^\top L Y^{(K)}] \equiv \sum_{(i,j)\in\mathcal{E}} \|Y_i^{(K)} - Y_j^{(K)}\|_2^2$. However, it is clear how to design $\ell(Y)$ such that minimizers do not degenerate in this way, e.g., by adding the first term in (2), or related generalizations, it has been previously established that oversmoothing is effectively mitigated, even while spreading information across the graph (Fu et al., 2023; Ma et al., 2021; Pan et al., 2020; Yang et al., 2021; Zhang et al., 2020; Zhu et al., 2021).

Unfolded GNNs provide other transparent entry points for customization as well. For example, if an energy function is insensitive to spurious graph edges, then a corresponding GNN architecture constructed via energy function descent is likely be robust against corrupted graphs from adversarial attacks or heterophily (Fu et al., 2023; Yang et al., 2021). More broadly, the flexibility of unfolded GNNs facilitates bespoke modifications of (3) for handling long-range dependencies (Xue et al., 2023), forming connections with the gradient dynamics of physical systems (Di Giovanni et al., 2023) and deep equilibrium models (Gu et al., 2020; Yang et al., 2022), exploiting the robustness of boosting algorithms (Sun et al., 2019), or differentiating the relative importance of features versus network effects in making predictions (Yoo et al., 2023).

## 2.3 Scalability Challenges and Candidate Solutions

As benchmarks continue to expand in size (see Table 1 below), it is no longer feasible to conduct full-graph GNN training using only a single GPU or even single machine, especially so for relatively deep unfolded GNNs. To address such GNN scalability challenges, there are presently two dominant lines of algorithmic workarounds. The first adopts various sampling techniques to extract much smaller computational subgraphs upon which GNN models can be trained in mini-batches. Relevant examples include neighbor sampling (Hamilton et al., 2017; Ying et al., 2018), layer-wise

sampling (Chen et al., 2018; Zou et al., 2019), and graph-wise sampling (Chiang et al., 2019; Zeng et al., 2021). For each of these, there exist both online and offline versions, where the former involves randomly sampling new subgraphs during each training epoch, while the latter (Zeng et al., 2021; Gasteiger et al., 2022) is predicated on a fixed set of subgraphs for all epochs; MuseGNN will ultimately be based on a novel integration and analysis of this approach as introduced in Section 3.

The second line of work exploits the reuse of historical embeddings, meaning the embeddings of nodes computed and saved during the previous training epoch. In doing so, much of the recursive forward and backward computations required for GNN training, as well as expensive memory access to high-dimensional node features, can be reduced (Chen et al., 2017; Fey et al., 2021; Huang et al., 2023). This technique has recently been applied to training unfolded GNN models (Xue et al., 2023), although available performance results do not cover large-scale graphs, there are no convergence guarantees, and no code is available (at least as of the time of this submission) for enabling further comparisons. Beyond this, we are unaware of prior work devoted to the scaling and coincident analysis of unfolded GNNs.

## 3 Graph-Regularized Energy Functions Infused with Sampling

Our goal of this section is to introduce a convenient family of energy functions formed by applying graph-based regularization to a set of subgraphs that have been sampled from a given graph of interest. For this purpose, we first present the offline sampling strategy which will undergird our approach, followed by details of energy functional form we construct on top of it for use with MuseGNN. Later, we conclude by analyzing special cases of these sampling-based energies, further elucidating relevant properties and connections with full-graph training.

### 3.1 Offline Sampling Foundation

In the present context, offline sampling refers to the case where we sample a fixed set of subgraphs from $\mathcal{G}$ once and store them, which can ultimately be viewed as a form of preprocessing step. More formally, we assume an operator $\Omega : \mathcal{G} \to \{\mathcal{G}_s\}_{s=1}^m$, where $\mathcal{G}_s = \{\mathcal{V}_s, \mathcal{E}_s\}$ is a subgraph of $\mathcal{G}$ containing $n_s = |\mathcal{V}_s|$ nodes, of which we assume $n'_s$ are target training nodes (indexed from 1 to $n'_s$), and $\mathcal{E}_s$ represents the edge set. The corresponding feature and label sets associated with the $s$-th subgraph are denoted $X_s \in \mathbb{R}^{n_s \times d}$ and $T_s \in \mathbb{R}^{n'_s \times d'}$, respectively.

There are several key reasons we employ offline sampling as the foundation for our scalable unfolded GNN architecture development. Firstly, conditioned on the availability of such pre-sampled/fixed subgraphs, there is no additional randomness when we use them to replace energy functions dependent on $\mathcal{G}$ and $X$ (e.g., as in (2)) with surrogates dependent on $\{\mathcal{G}_s, X_s\}_{s=1}^m$. Hence we retain a deterministic energy substructure contributing to a more transparent bilevel (upper- and lower-level) optimization process and attendant node-wise embeddings that serve as minimizers. Secondly, offline sampling allows us to conduct formal convergence analysis that is agnostic to the particular sampling operator $\Omega$. In this way, we need not compromise flexibility in choosing a practically-relevant $\Omega$ in order to maintain desirable convergence guarantees. For example, as will be detailed later, ShadowKHop (Zeng et al., 2021) sampling pairs well with our approach and directly adheres to our convergence analysis in Section 5. And lastly, offline sampling facilitates an attractive balance between model accuracy and efficiency within the confines of unfolded GNN architectures. As will be shown in Section 6, we can match the accuracy of full-graph training with an epoch time similar to online sampling methods, e.g., neighbor sampling.

### 3.2 Energy Function Formulation

To integrate offline sampling into a suitable graph-regularized energy function, we first introduce two sets of auxiliary variables that will serve as more flexible input arguments. Firstly, to accommodate multiple different embeddings for the same node appearing in multiple subgraphs, we define $Y_s \in \mathbb{R}^{n_s \times d}$ for each subgraph index $s = 1, \ldots, m$, as well as $\mathbb{Y} = \{Y_s\}_{s=1}^m \in \mathbb{R}^{(\sum n_s) \times d}$ to describe the concatenated set. Secondly, we require additional latent variables that, as we will later see, facilitate a form of controllable linkage between the multiple embeddings that may exist for a given node (i.e., when a given node appears in multiple subgraphs). For this purpose, we define the latent variables

as $M \in \mathbb{R}^{n \times d}$, where each row can be viewed as a shared summary embedding associated with each node in the original/full graph.

We then define our sampling-based extension of (2) for MuseGNN as

$$\ell_{\text{muse}}(\mathbb{Y}, M) := \sum_{s=1}^{m} \left[ \|Y_s - f(X_s; W)\|_F^2 + \lambda \operatorname{tr}(Y_s^\top L_s Y_s) + \gamma \|Y_s - \mu_s\|_F^2 + \sum_{i=1}^{n_s} \zeta(Y_{s,i}) \right], \quad (4)$$

where $L_s$ is the graph Laplacian associated with $\mathcal{G}_s$, $\gamma \geq 0$ controls the weight of the additional penalty factor, and each $\mu_s \in \mathbb{R}^{n_s \times d}$ is derived from $M$ as follows. Let $I(s, i)$ denote a function that maps the index of the $i$-th node in subgraph $s$ to the corresponding node index in the full graph. For each subgraph $s$, we then define $\mu_s$ such that its $i$-th row satisfies $\mu_{s,i} = M_{I(s,i)}$; per this construction, $\mu_{s,i} = \mu_{s',j}$ if $I(s, i) = I(s', j)$. Consequently, $\{\mu_s\}_{s=1}^{m}$ and $M$ represent the same overall set of latent embeddings, whereby the former is composed of repeated samples from the latter aligned with each subgraph.

Overall, there are three prominent factors which differentiate (4) from (2):

1. $\ell_{\text{muse}}(\mathbb{Y}, M)$ involves a deterministic summation over a fixed set of sampled subgraphs, where each $Y_s$ is unique while $W$ (and elements of $M$) are shared across subgraphs.

2. Unlike (2), the revised energy involves both an expanded set of node-wise embeddings $\mathbb{Y}$ as well as auxiliary summary embeddings $M$. When later forming GNN layers designed to minimize $\ell_{\text{muse}}(\mathbb{Y}, M)$, we must efficiently optimize over *all* of these quantities, which alters the form of the final architecture.

3. The additional $\gamma \|Y_s - \mu_s\|_F^2$ penalty factor acts to enforce dependencies between the embeddings of a given node spread across different subgraphs.

With respect to the latter, it is elucidating to consider minimization of $\ell_{\text{muse}}(\mathbb{Y}, M)$ over $\mathbb{Y}$ with $M$ set to some fixed $M'$. In this case, up to an irrelevant global scaling factor and additive constant, the energy can be equivalently reexpressed as

$$\ell_{\text{muse}}(\mathbb{Y}, M = M') \equiv \sum_{s=1}^{m} \left[ \|Y_s - [f'(X_s; W) + \gamma' \mu_s]\|_F^2 + \lambda' \operatorname{tr}(Y_s^\top L_s Y_s) + \sum_{i=1}^{n_s} \zeta'(Y_{s,i}) \right], \quad (5)$$

where $f'(X_s; W) := \frac{1}{1+\gamma} f(X_s; W)$, $\gamma' := \frac{\gamma}{1+\gamma}$, $\lambda' := \frac{\lambda}{1+\gamma}$, and $\zeta'(Y_{s,i}) := \frac{1}{1+\gamma} \zeta(Y_{s,i})$. From this expression, we observe that, beyond inconsequential rescalings (which can be trivially neutralized by simply rescaling the original choices for $\{f, \gamma, \lambda, \zeta\}$), the role of $\mu_s$ is to refine the initial base predictor $f(X_s; W)$ with a corrective factor reflecting embedding summaries from other subgraphs sharing the same nodes. Conversely, when we minimize $\ell_{\text{muse}}(\mathbb{Y}, M)$ over $M$ with $\mathbb{Y}$ fixed, we find that the optimal $\mu_s$ for every $s$ is equal to the *mean* embedding for each constituent node across all subgraphs. Hence $M'$ chosen in this way via alternating minimization has a natural interpretation as grounding each $Y_s$ to a shared average representation reflecting the full graph structure. We now consider two limiting special cases of $\ell_{\text{muse}}(\mathbb{Y}, M)$ that provide complementary contextualization.

## 3.3 NOTABLE LIMITING CASES

We first consider setting $\gamma = 0$, in which case the resulting energy completely decouples across each subgraph such that we can optimize each

$$\ell_{\text{muse}}^s(Y_s) := \|Y_s - f(X_s; W)\|_F^2 + \lambda \operatorname{tr}(Y_s^\top L_s Y_s) + \sum_{i=1}^{n_s} \zeta(Y_{s,i}) \quad \forall s \quad (6)$$

independently. Under such conditions, the only cross-subgraph dependency stems from the shared base model weights $W$ which are jointly trained. Hence $\ell_{\text{muse}}^s(Y_s)$ is analogous to a full graph energy as in (2) with $\mathcal{G}$ replaced by $\mathcal{G}_s$. In Section 5 we will examine convergence conditions for the full bilevel optimization process over all $Y_s$ and $W$ that follows from the $\gamma = 0$ assumption. We have also found that this simplified setting performs well in practice; see Appendix B.1 for ablations.

At the opposite extreme when $\gamma = \infty$, we are effectively enforcing the constraint $Y_s = \mu_s$ for all $s$. As such, we can directly optimize all $Y_s$ out of the model leading to the reduced $M$-dependent energy

$$\ell_{\text{muse}}(M) := \sum_{s=1}^{m} \left[ \|\mu_s - f(X_s; W)\|_F^2 + \lambda \operatorname{tr}(\mu_s^\top L_s \mu_s) + \sum_{i=1}^{n_s} \zeta(\mu_{s,i}) \right]. \tag{7}$$

Per the definition of $\{\mu_s\}_{s=1}^m$ and the correspondence with unique node-wise elements of $M$, this scenario has a much closer resemblance to full graph training with the original $\mathcal{G}$. In this regard, as long as a node from the original graph appears in at least one subgraph, then it will have a single, unique embedding in (7) aligned with a row of $M$.

Moreover, we can strengthen the correspondence with full-graph training via the suitable selection of the offline sampling operator $\Omega$. In fact, there exists a simple uniform sampling procedure such that, at least in expectation, the energy from (7) is equivalent to the original full-graph version from (2), with the role of $M$ equating to $Y$. More concretely, we present the following (all proofs are deferred to Appendix D):

**Proposition 3.1.** *Suppose we have $m$ subgraphs $(\mathcal{V}_1, \mathcal{E}_1), \ldots, (\mathcal{V}_m, \mathcal{E}_m)$ constructed independently such that $\forall s = 1, \ldots, m, \forall u, v \in \mathcal{V}, \Pr[v \in \mathcal{V}_s] = \Pr[v \in \mathcal{V}_s \mid u \in \mathcal{V}_s] = p; (i,j) \in \mathcal{E}_s \iff i \in \mathcal{V}_s, j \in \mathcal{V}_s, (i,j) \in \mathcal{E}$. Then when $\gamma = \infty$, we have $\mathbb{E}[\ell_{muse}(M)] = mp\, \ell(M)$ with the $\lambda$ in $\ell(M)$ rescaled to $p\lambda$.*

Strengthened by this result, we can more directly see that (7) provides an intuitive bridge between full-graph models based on (2) and subsequent subgraph models we intend to build via (4), with the later serving as an unbiased estimator of the former. Even so, we have found that relatively small $\gamma$ values nonetheless work well in practice (although in principle it is actually possible to handle $\gamma = \infty$ using techniques like ADMM (Boyd et al., 2011)).

## 4  FROM SAMPLING-BASED ENERGIES TO THE MUSEGNN FRAMEWORK

Having defined and motivated a family of sampling-based energy functions vis-a-vis (4), we now proceed to derive minimization steps that will serve as GNN model layers that define MuseGNN forward and backward passes as summarized in Algorithm 1. Given that there are two input arguments, namely $\mathbb{Y}$ and $M$, it is natural to adopt an alternating minimization strategy whereby we fix one and optimize over the other, and vice versa.

With this in mind, we first consider optimization over $\mathbb{Y}$ with $M$ fixed. Per the discussion in Section 3.2, when conditioned on a fixed $M$, $\ell_{\text{muse}}(\mathbb{Y}, M)$ decouples over subgraphs. Consequently, optimization can proceed using subgraph-independent proximal gradient descent, leading to the update rule

$$Y_s^{(k)} = \operatorname{prox}_\zeta \left[ Y_s^{(k-1)} - \alpha \left( [(1 + \gamma)I + \lambda L_s] Y_s^{(k-1)} - [f(X_s; W) + \gamma \mu_s] \right) \right], \quad \forall s. \tag{8}$$

Here the input argument to the proximal operator is given by a gradient step along (4) w.r.t. $Y_s$. We also remark that execution of (8) over $K$ iterations represents the primary component of a single forward training pass of our proposed MuseGNN framework as depicted on lines 6-8 of Algorithm 1.

We next turn to optimization over $M$ which, as mentioned previously, can be minimized by the mean of the subgraph-specific embeddings for each node. However, directly computing these means is problematic for computational reasons, as for a given node $v$ this would require the infeasible collection of embeddings from all subgraphs containing $v$. Instead, we adopt an online mean estimator with forgetting factor $\rho$. For each node $v$ in the full graph, we maintain a mean embedding $M_v$ and a counter $c_v$. When this node appears in the $s$-th subgraph as node $i$ (where $i$ is the index within the subgraph), we update the mean embedding and counter via

$$M_{I(s,i)} \leftarrow \frac{\rho c_{I(s,i)}}{c_{I(s,i)} + 1} M_{I(s,i)} + \frac{(1 - \rho) c_{I(s,i)} + 1}{c_{I(s,i)} + 1} Y_{s,i}^{(K)}, \qquad c_{I(s,i)} \leftarrow c_{I(s,i)} + 1. \tag{9}$$

Also shown on line 9 of Algorithm 1, we update $M$ and $c$ once per forward pass, which serves to refine the effective energy function observed by the core node embedding updates from (8).

For the backward pass, we compute gradients of

$$\mathcal{L}_{\text{muse}}(W, \theta) := \sum_{s=1}^{m} \sum_{i=1}^{n_s'} \mathcal{D}\left( g\left[ Y_s^{(K)}(W)_i; \theta \right], T_{s,i} \right) \tag{10}$$

w.r.t. $W$ and $\theta$ as listed on line 10 of Algorithm 1, where $\mathcal{L}_{\text{muse}}(W, \theta)$ is a sampling-based modification of (1). For $W$ though, we only pass gradients through the calculation of $Y_s^{(K)}$, not the full online $M$ update; however, provided $\rho$ is chosen to be sufficiently large, $M$ will change slowly relative to $Y_s^{(K)}$ such that this added complexity is not necessary for obtaining reasonable convergence.

---

**Algorithm 1** Training procedure for MuseGNN

> **Input**: $\{\mathcal{G}_s\}_{s=1}^m$: subgraphs, $\{X_s\}_{s=1}^m$: features, $K$: # unfolded layers, $E$: # epochs
> 1: Randomly initialize $W$ and $\theta$; initialize $c \in \mathbb{R}^n$ and $M \in \mathbb{R}^{n \times d}$ to be zero
> 2: **for** $e = 1, 2, \ldots, E$ **do**
> 3:     **for** $s = 1, 2, \ldots, m$ **do**
> 4:         $\mu_{s,i} \leftarrow M_{I(s,i)}, i = 1, 2, \ldots, n_s$
> 5:         $Y_s^{(0)} \leftarrow f(X_s; W)$
> 6:         **for** $k = 1, 2, \ldots, K$ **do**
> 7:             Update $Y_s^{(k)}$ from $Y_s^{(k-1)}$ by (8) using $\mu_s$
> 8:         **end for**
> 9:         Update $M$ and $c$ using (9) with $Y_{s,i}^{(K)}, i = 1, 2, \ldots, n_s$
> 10:        Update $W$ and $\theta$ via SGD over $\mathcal{L}_{\text{muse}}$ from (10)
> 11:     **end for**
> 12: **end for**

---

## 5   Convergence Analysis of MuseGNN

**Global Convergence with $\gamma = 0$.** In the more restrictive setting where $\gamma = 0$, we derive conditions whereby the entire bilevel optimization pipeline converges to a solution that jointly minimizes both lower- and upper-level energy functions in a precise sense to be described shortly. We remark here that establishing convergence is generally harder for bilevel optimization problems relative to more mainstream, single-level alternatives (Colson et al., 2005). To describe our main result, we first require the following definition:

**Definition 5.1.** Assume $f(X; W) = XW$, $\gamma = 0$, $\zeta(y) = 0$, $g(y; \theta) = y$, and that $\mathcal{D}$ is a Lipschitz continuous convex function. Given the above, we then define $\mathcal{L}_{\text{muse}}^{(k)}(W)$ as (10) with $K$ set to $k$. Analogously, we also define $\mathcal{L}_{\text{muse}}^*(W)$ as (10) with $Y_s^{(K)}$ replaced by $Y_s^* := \arg\min_{Y_s} \ell_{\text{muse}}^s(Y_s)$ for all $s$.

**Theorem 5.2.** *Let $W^*$ be the optimal value of the loss $\mathcal{L}_{muse}^*(W)$ per Definition 5.1, while $W^{(t)}$ denotes the value of $W$ after $t$ steps of stochastic gradient descent over $\mathcal{L}_{muse}^{(k)}(W)$ with diminishing step sizes $\eta_t = O(\frac{1}{\sqrt{t}})$. Then provided we choose $\alpha \in \left(0, \min_s \|I + \lambda L_s\|_2^{-1}\right]$ and $Y_s^{(0)} = f(X_s; W)$, for some constant $C$ we have that*

$$\mathbb{E}\left[\mathcal{L}_{muse}^{(k)}(W^{(t)})\right] - \mathcal{L}_{muse}^*(W^*) \leq O\left(\frac{1}{\sqrt{t}} + e^{-Ck}\right).$$

Given that $\mathcal{L}_{\text{muse}}^*(W^*)$ is the global minimum of the combined bilevel system, this result guarantees that we can converge arbitrarily close to it with adequate upper- and lower-level iterations, i.e., $t$ and $k$ respectively. Note also that *we have made no assumption on the sampling method*. In fact, as long as offline sampling is used, convergence is guaranteed, although the particular offline sampling approach can potentially impact the convergence rate; see Appendix C.2 for further details.

**Lower-Level Convergence with arbitrary $\gamma$.** We now address the more general scenario where $\gamma \geq 0$ is arbitrary. However, because of the added challenge involved in accounting for alternating minimization over both $\mathbb{Y}$ and $M$, it is only possible to establish conditions whereby the lower-level energy (4) is guaranteed to converge as follows.

**Theorem 5.3.** *Assume $\zeta(y) = 0$. Suppose that we have a series of $\mathbb{Y}^{(k)}$ and $M^{(k)}$, $k = 0, 1, 2, \ldots$ constructed following the updating rules $\mathbb{Y}^{(k)} := \arg\min_Y \ell_{muse}(\mathbb{Y}, M^{(k-1)})$ and $M^{(k)} := \arg\min_M \ell_{muse}(\mathbb{Y}^{(k)}, M)$, with $\mathbb{Y}^{(0)}$ and $M^{(0)}$ initialized arbitrarily. Then*

$$\lim_{k \to \infty} \ell_{muse}(\mathbb{Y}^{(k)}, M^{(k)}) = \inf_{\mathbb{Y}, M} \ell_{muse}(\mathbb{Y}, M). \tag{11}$$

While technically this result does not account for minimization over the upper-level MuseGNN loss, we still empirically find that the entire process outlined by Algorithm 1, including the online mean update, is nonetheless able to converge in practice. See Appendix C.1 for an illustration.

## 6 EXPERIMENTS

We now seek to empirically show that MuseGNN serves as a reliable unfolded GNN model that:

1. Preserves competitive accuracy across datasets of widely varying size, including the very largest publicly-available graph benchmarks, with a single fixed architecture, and

2. Operates with comparable computational complexity relative to common alternatives that are also capable of scaling to the largest graphs.

For context though, we note that graph benchmark leaderboards often include top-performing entries based on complex compositions of existing models and training tricks, at times dependent on additional features or external data not included in the original designated dataset. Although these approaches have merit, our goal herein is not to compete with them, as they typically vary from dataset to dataset. Additionally, they are more frequently applied to smaller graphs using architectures that have yet to be consistently validated across multiple, truly large-scale benchmarks.

**Datasets.** We evaluate the performance of MuseGNN on node classification tasks from the Open Graph Benchmark (OGB) (Hu et al., 2020; 2021) and the Illinois Graph Benchmark (IGB) (Khatua et al., 2023). Table 1 presents the details of these datasets, which are based on homogeneous graphs spanning a wide range of sizes. Of particular note is `IGB-full`, currently the largest publicly-available graph benchmark, along with a much smaller version called `IGB-tiny` that we include for comparison purposes. We also point out that while `MAG240M` is originally a heterogenous graph, we follow the common practice of homogenizing it (Hu et al., 2021); similarly for IGB datasets, we adopt the provided homogeneous versions.

Table 1: Dataset details, including node feature dimension (Dim.) and number of classes (# Class).

| Dataset | $|\mathcal{V}|$ | $|\mathcal{E}|$ | Dim. | # Class | Dataset Size |
|---|---|---|---|---|---|
| `ogbn-arxiv` (Hu et al., 2020) | 0.17M | 1.2M | 128 | 40 | 182MB |
| `IGB-tiny` (Khatua et al., 2023) | 0.1M | 0.5M | 1024 | 19 | 400MB |
| `ogbn-products` (Hu et al., 2020) | 2.4M | 123M | 100 | 47 | 1.4GB |
| `ogbn-papers100M` (Hu et al., 2020) | 111M | 1.6B | 128 | 172 | 57GB |
| `MAG240M` (Hu et al., 2021) | 244.2M | 1.7B | 768 | 153 | 377GB[1] |
| `IGB-full` (Khatua et al., 2023) | 269.3M | 4.0B | 1024 | 19 | 1.15TB |

**MuseGNN Design.** For *all* experiments, we choose the following fixed MuseGNN settings: Both $f(X; W)$ and $g(Y; \theta)$ are 3-layer MLPs, the number of unfolded layers $K$ is 8 (rationale discussed later below), and the embedding dimension $d$ is 512, and the forgetting factor $\rho$ for the online mean estimation is 0.9. For offline subgraph sampling, we choose a variation on neighbor sampling called ShadowKHop (Zeng et al., 2021), which loosely approximates the conditions of Proposition 3.1. See Appendix A for further details regarding the MuseGNN implementation and hyperparameters.

**Baseline Models.** With the stated objectives of this section in mind, we compare MuseGNN with GCN (Kipf & Welling, 2017), GAT (Velickovic et al., 2018), and GraphSAGE (Hamilton et al., 2017), in each case testing with both neighbor sampling (NS) (Hamilton et al., 2017) and GNNAutoScale (GAS) (Fey et al., 2021) for scaling to large graphs. As a point of reference, we note that GAT with neighbor sampling is currently the top-performing homogeneous graph model on both the `MAG240M` and `IGB-full` leaderboards. We also compare with an analogous full-graph (FG) unfolded GNN (UGNN) with the same architecture as MuseGNN but without scalable sampling.

**Accuracy Comparisons.** As shown in Table 2, MuseGNN achieves similar accuracy to a comparable full-graph unfolded GNN model on the small datasets (satisfying our first objective from

---

[1]The size includes author and institution features.

above), while the latter is incapable of scaling to even the mid-sized `ogbn-products` benchmark. Meanwhile, MuseGNN is generally better than the other GNN baselines, particularly so on the largest dataset, `IGB-full`. Notably, MuseGNN exceeds the performance of GAT with neighbor sampling, the current SOTA for homogeneous graph models on both `MAG240M` and `IGB-full`. Please also refer to Appendix B.1 for MuseGNN accuracy ablations involving $\gamma$, where we show that while even $\gamma = 0$ achieves strong results, increasing $\gamma > 0$ can lead to further improvement. Likewise, Appendix B.2 contains experiments showing that the improvement of MuseGNN is not merely a product of using ShadowKHop sampling instead of more traditional neighbor sampling with baseline models; there we show that when switched to offline ShadowKHop sampling, the baseline GNNs degrade further, possibly because they are not anchored to an integrated energy.

**Timing Comparisons.** Turning to model complexity, Table 3 displays the training speed of MuseGNN relative to two common baselines. From these results, we observe that MuseGNN executes with a similar epoch time (satisfying our second objective from above).

Table 2: Node classification accuracy (%) on the test set, except for `MAG240M` which only has labels for validation set. Bold numbers denote the highest accuracy, and error bars are missing for some GNN baselines that were too expensive to compute. For the two largest datasets, MuseGNN is currently the top-performing homogeneous graph model on the relevant OGB-LSC and IGB leaderboards respectively, even while maintaining the attractive attributes of an unfolded GNN. We omit error bars for baseline results in the two largest datasets because of the high cost to run them.

| | | Small | | Medium | Large | | Largest | |
|---|---|---|---|---|---|---|---|---|
| **Model** | | arxiv | IGB-tiny | products | papers100M | | MAG240M | IGB-full |
| GCN (NS) | | 69.71±0.25 | 69.77±0.11 | 78.49±0.53 | 65.83±0.36 | | 65.24 | 48.59 |
| SAGE (NS) | | 70.49±0.20 | 72.42±0.09 | 78.29±0.16 | 66.20±0.13 | | 66.79 | 54.95 |
| GAT (NS) | | 69.94±0.28 | 69.70±0.09 | 79.45±0.59 | 66.28±0.08 | | 67.15 | 55.51 |
| GCN (GAS) | | 71.68±0.3 | 67.86±0.20 | 76.6±0.3 | 54.2±0.7 | | OOM | OOM |
| SAGE (GAS) | | 71.35±0.4 | 69.35±0.06 | 77.7±0.7 | 57.9±0.4 | | OOM | OOM |
| GAT (GAS) | | 70.89±0.1 | 69.23±0.17 | 76.9±0.5 | OOM | | OOM | OOM |
| UGNN (FG) | | **72.74±0.25** | 72.44±0.09 | OOM | OOM | | OOM | OOM |
| MuseGNN | | 72.50±0.19 | **72.80±0.02** | **81.23±0.39** | **66.82±0.02** | | **67.26±0.06** | **60.21±0.18** |

Table 3: Training speed (epoch time) in seconds; hardware configurations in Appendix A. Even on the largest graph, `IGB-full`, MuseGNN remains comparable to popular baselines that adopt online sampling.

| **Model** | papers-100M | MAG240M | IGB-full |
|---|---|---|---|
| SAGE (NS) | 102.48 | 795.19 | 17279.98 |
| GAT (NS) | 164.14 | 1111.30 | 19789.79 |
| MuseGNN | 158.85 | 1370.47 | 20413.69 |

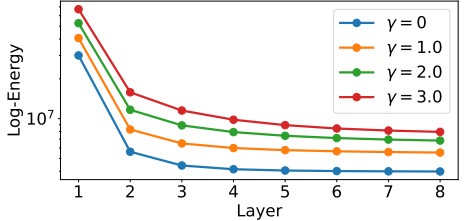

Figure 1: Convergence of (4) for MuseGNN trained on `papers100M` with varying $\gamma$.

**Convergence Illustration.** In Figure 1 we show the empirical convergence of (4) w.r.t. $\mathbb{Y}$ during the forward pass of MuseGNN models on `ogbn-papers100M` for differing values of $\gamma$. In all cases the energy converges within 8 iterations, supporting our choice of $K = 8$ for experiments.

## 7 CONCLUSION

In this work, we have proposed MuseGNN, an unfolded GNN model that scales to large datasets by incorporating offline graph sampling into the design of its lower-level, architecture-inducing energy function. In so doing, MuseGNN readily handles graphs with $\sim 10^8$ or more nodes and high-dimensional node features, exceeding 1TB in total size, all while maintaining transparent layers inductive biases and concomitant convergence guarantees.

ETHICS STATEMENT

While we do not envision that there are any noteworthy risks introduced by our work, it is of course always possible that a large-scale model like MuseGNN could be deployed, either intentionally or unintentionally, in such a way as to cause societal harm. For example, a graph neural network designed to screen fraudulent credit applications could be biased against certain minority groups.

REPRODUCIBILITY STATEMENT

We present the detailed algorithm for training MuseGNN in Section 4. Meanwhile, supplementary experimental details are recorded in Appendix A and proofs for all technical results are deferred to Appendix D. Additionally, because ICLR submissions are immediately public, we choose not to release the model code until the decision.

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

## A  EXPERIMENT DETAILS

**Model Details.**  For MuseGNN, we choose the base model $f(X; W)$ and output function $g(Y; \theta)$ to be two shallow 3-layer MLPs with standard residual skip connections (although for the somewhat differently-structured `ogbn-products` data we found that skip connections were not necessary). The number of unfolded layers, $K$, is selected to be 8, and the hidden dimension $d$ is set to 512. The forgetting factor $\rho$ for the online mean estimation is set to be 0.9. All the models and experiments are implemented in PyTorch (Paszke et al., 2019) using the Deep Graph Library (DGL) (Wang, 2019).

**Offline Samples.**  As supported by Proposition 3.1, subgraphs induced from nodes (where all the edges between sampled nodes are kept) pair naturally with the energy function. Therefore we use node-induced subgraphs for our experiments. For `ogbn-arxiv`, we use the 2-hop full neighbor because it is a relatively small graph. For `IGB-tiny`, `ogbn-papers100M`, and `MAG240M`, we use ShadowKHop with fanout [5,10,15]. For the rest of the datasets, we use ShadowKHop with fanout [10, 15]. Note that ShadowKHop sampler performs node-wise neighbor sampling and returns the subgraph induced by all the sampled nodes.

**Training Hyperparameters.**  In the training process, we set the dropout rate of the MLP layers to be 0.2, and we do not have dropout between the propagation layers. The parameters are optimized by Adam optimizer (Kingma & Ba, 2014) with the weight decay parameter set to 0 and the learning rate being 0.001. For `ogbn-arxiv` and `ogbn-papers100M`, $\alpha = 0.05, \lambda = 20$. For `MAG240M` and the `IGB` series datasets, $\alpha = 0.2, \lambda = 4$. And for `ogbn-products`, $\alpha = \lambda = 1$ with preconditioning on the degree matrix for each unfolded step/layer. The batch size $n'_s$ in the offline samples are all set to 1000. For large-scale datasets, an expensive hyperparameter grid search as commonly used for GNN tuning is not feasible. Hence we merely applied simple heuristics informed from training smaller models to pick hyperparameters for the larger datasets. Also it is worth noticing that $\alpha$ should not in principle affect accuracy if set small enough, since the model will eventually converge. In this regard, $\alpha$ can be set dependent on $\lambda$, since the latter determines the size of $\alpha$ needed for convergence.

**Evaluation Process.**  In the presented results, the evaluation process for validation and test datasets uses the same pipeline as the training process does: doing offline sampling first and then use these fixed samples to do the calculation. The only difference is that in the evaluation process, the backward propagation is not performed. While we used the same pipeline for training and testing, MuseGNN is modular and we could optionally use online sampling for the evaluation or even multi-hop full-neighbor loader in the evaluation time.

**Configurations in the Speed Experiments.**  We use a single AWS EC2 p4d.24xlarge instance to run the speed experiments. It comes with dual Intel Xeon Platinum 8275CL CPU (48 cores, 96 threads), 1.1TB main memory and 8 A100 (40GB) GPUs. All the experiments are run on single GPU. When running MuseGNN, the pre-sampled graph structures are stored in the NVMe SSDs and the input feature is loaded into the main memory (for `IGB-full`, its feature exceeds the capacity of the main memory, so mmap is employed). In the baseline setting, `ogbn-papers100M` and `MAG240M` are paired with neighbor sampling of fanout [5, 10, 15] and `IGB-full` is paired with neighbor sampling of fanout [10, 15]. So the fanout for the online neighbor sampling baselines and MuseGNN are the same so that they can have a fair comparison. In these experiments, the hidden size is set to 256 as commonly does.

**Model Code.**  Because ICLR submissions are public, we choose not to release the code until the decision.

**Extra Dataset.**  Table 4 specifies details of the extra dataset we used for the ablation study in Appendix B.1.

Table 4: Dataset details, including node feature dimension (Dim.) and number of classes (# Class).

| Dataset | $|\mathcal{V}|$ | $|\mathcal{E}|$ | Dim. | # Class | Dataset Size |
|---|---|---|---|---|---|
| IGB-medium (Khatua et al., 2023) | 10M | 120M | 1024 | 19 | 40.8GB |

# B ADDTIONAL EXPERIMENTS AND DISCUSSIONS

## B.1 FURTHER DETAILS REGARDING THE ROLE OF $\gamma$ IN MUSEGNN

The incorporation of $\gamma > 0$ serves two purposes within our MuseGNN framework and its supporting analysis. First, by varying $\gamma$ we are able to build a conceptual bridge between the decoupled/inductive $\gamma = 0$ base scenario, and full-graph training as $\gamma \to \infty$, provided the sampled subgraphs adhere to the conditions of Proposition 3.1. In this way, the generality of $\gamma > 0$ has value in terms of elucidating connections between modeling regimes, independently of empirical performance.

That being said, the second role is more pragmatic, as $\gamma > 0$ can indeed improve predictive accuracy. As shown in the ablation in Table 5 (where all hyperparameters except $\gamma$ remain fixed), $\gamma = 0$ already achieves good performance; however, increasing $\gamma$ leads to further improvements. This is likely because larger $\gamma$ enables the model to learn similar embeddings for the same node in different subgraphs, which may be more robust to the sampling process.

Table 5: Ablation study of the penalty factor $\gamma$. Results shown represent the accuracy (%) on the test set. Bold numbers denote the best performing method.

| | $\gamma = 0$ | $\gamma = 0.1$ | $\gamma = 0.5$ | $\gamma = 1$ | $\gamma = 2$ | $\gamma = 3$ |
|---|---|---|---|---|---|---|
| papers100M | 66.29 | 66.31 | 66.53 | 66.45 | 66.64 | **66.82** |
| IGB-tiny | 72.66 | 72.78 | **72.81** | N/A | N/A | N/A |
| IGB-medium | 75.18 | 75.80 | **75.83** | N/A | N/A | N/A |
| ogbn-products | 80.42 | 80.92 | **81.23** | N/A | N/A | N/A |

That being said, while we can improve accuracy with $\gamma > 0$, this generality comes with an additional memory cost for storing the required mean vectors. However, this memory complexity for the mean vectors is only $O(nd)$, which is preferable to the memory complexity $O(ndK)$ of GAS (Fey et al., 2021), which depends on the number of layers $K$. Importantly, this more modest $O(nd)$ storage is only an upper bound for the MuseGNN memory complexity, as the $\gamma = 0$ case is already very effective even without requiring additional storage. As an additional point of reference, the GAS-related approach from (Xue et al., 2023) also operates with only single-layer $O(nd)$ of additional storage; however, unlike MuseGNN, this storage is mandatory for large graphs (not amenable to full-graph training) and therefore serves as a *lower* bound for memory complexity. Even so, the approach from (Xue et al., 2023) is valuable and complementary, and should their code become available in the future, it would be interesting to explore the possibility of integrating with MuseGNN.

## B.2 SAMPLING ABLATION ON THE BASELINE MODELS

We choose neighbor sampling for the baselines because it is most commonly used, but since ShadowKHop is paired with MuseGNN, we use Table 6 to show that the improvement is not coming from the change in the sampling method but the usage of energy-based scalable unfolded model. In Table 6, the baseline models are trained on ogbn-papers100M with the same offline ShadowKHop samples used in the training of MuseGNN, but they suffer from a decrease in the accuracy compared with the online neighbor sampling counterparts, so the change in sampling method cannot account for the enhancement in the accuracy.

## B.3 ALTERNATIVE SAMPLING METHOD FOR MUSEGNN

We also paired MuseGNN with online neighbor sampling, the most popular choice for sampling, on ogbn-papers100M to account for our design choice of sampling methods coupled with

Table 6: Ablation study of the accuracy improvement compared with baselines. Shown results are the accuracy (%) on the test set of `ogbn-papers100M`.

|  | Baseline Model | | |
| --- | --- | --- | --- |
| Sampling Method | GCN | GAT | SAGE |
| ShadowKHop (offline) | 64.89 | 63.84 | 65.57 |
| Neighbor Sampling (online) | 65.83 | 66.28 | 66.2 |

MuseGNN. Theoretically, the convergence guarantee no longer holds true for online sampling. Additionally, neighbor sampling removes terms regarding edge information from the energy function, so we expect it to be worse than the offline ShadowKHop sampling that we use. This is also supported by Proposition 3.1 because neighbor sampling does not provide subgraphs induced from the nodes with all the edges. Empirically, with online neighbor sampling, the test accuracy is 66.11%; it is indeed lower than the 66.82% from our offline ShadowKHop results.

## C  ADDITIONAL CONVERGENCE DETAILS

### C.1  EMPERICAL CONVERGENCE RESULTS

Here we show the empirical results of the previous convergence analysis. Though we do not have the full bilevel convergence results for the $\gamma > 0$ case like Theorem 5.2, Figure 2 shows that in real-world dataset, the bilevel optimization system will still converge with the imprecise estimation of optimal embeddings $Y_s^{(K)} \approx Y_s^*$ and the online mean estimation of $M$.

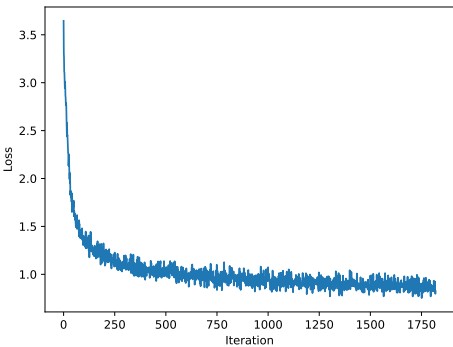

Figure 2: Convergence of the upper-level loss on `ogbn-arxiv` dataset for 20 epochs with penalty factor $\gamma = 1$.

### C.2  A NOTE ON SAMPLING METHODS IMPACTING CONVERGENCE RATES

As stated in the main paper, offline sampling allows us to conduct formal convergence analysis that is agnostic to the particular sampling operator. So our convergence guarantee is valid for all sampling methods. However, different sampling methods can still actually have an impact on the convergence *rate* of the bilevel optimization. This is because different sampling methods will produce subgraphs with different graph Laplacian matrices, and the latter may have differing condition numbers $\frac{\sigma}{\tau}$ as mentioned later in Lemma D.4. In this regard, a larger such condition number will generally yield a slower convergence rate. Further details are elaborated in the proof of Theorem 5.2 in Appendix D.2.

# D PROOFS

## D.1 CONNECTION WITH FULL GRAPH TRAINING

**Proposition 3.1.** *Suppose we have $m$ subgraphs $(\mathcal{V}_1, \mathcal{E}_1), \dots, (\mathcal{V}_m, \mathcal{E}_m)$ constructed independently such that $\forall s = 1, \dots, m, \forall u, v \in \mathcal{V}, \Pr[v \in \mathcal{V}_s] = \Pr[v \in \mathcal{V}_s \mid u \in \mathcal{V}_s] = p; (i,j) \in \mathcal{E}_s \iff i \in \mathcal{V}_s, j \in \mathcal{V}_s, (i,j) \in \mathcal{E}$. Then when $\gamma = \infty$, we have $\mathbb{E}[\ell_{muse}(M)] = mp\, \ell(M)$ with the $\lambda$ in $\ell(M)$ rescaled to $p\lambda$.*

*Proof.* The probability that an edge is sampled is $\Pr[(i,j) \in \mathcal{E}_s \mid (i,j) \in \mathcal{E}] = p^2$. $\forall v \in \mathcal{V}$, if $v \in \mathcal{V}_s, v \in \mathcal{V}_t$, then $Y_{s,v} = Y_{t,v} = M_v$. We have

$$\ell_{\text{muse}}(M) = \sum_{s=1}^{m} \left[ \sum_{v \in \mathcal{V}_s} \left( \|M_v - f(X; W)_v\|_2^2 + \zeta(M_v) \right) + \frac{\lambda}{2} \sum_{(i,j) \in \mathcal{E}_s} \|M_i - M_j\|_2^2 \right]$$

$$\mathbb{E}\left[\ell_{\text{muse}}(M)\right] = \sum_{s=1}^{m} \sum_{v \in \mathcal{V}_s} \left( \|M_v - f(X; W)_v\|_2^2 + \zeta(M_v) \right) \cdot \Pr[v \in \mathcal{V}_s]$$

$$+ \frac{\lambda}{2} \sum_{s=1}^{m} \sum_{(i,j) \in \mathcal{E}_s} \|M_i - M_j\|_2^2 \cdot \Pr[(i,j) \in \mathcal{E}_s]$$

$$= mp \sum_{v \in \mathcal{V}} \left( \|M_v - f(X; W)_v\|_2^2 + \zeta(M_v) \right) + mp^2 \frac{\lambda}{2} \sum_{(i,j) \in \mathcal{E}} \|M_i - M_j\|_2^2$$

$$= mp \left[ \|M - f(X; W)\|_F^2 + p\lambda \operatorname{tr}\left(M^\top L M\right) + \sum_{i=1}^{n} \zeta(M_i) \right]$$

$$= mp\, \ell(M)$$

$\square$

## D.2 FULL CONVERGENCE ANALYSIS

According to Definition 5.1, $Y_s^*(W) = (I + \lambda L_s)^{-1} f(X_s; W) := P_s^* f(X_s; W)$ is the optimal embedding for the subgraph energy $\ell_{\text{muse}}^s(Y_s)$. Additionally, we initialize $Y_s^{(0)}$ as $f(X_s; W)$, so $Y_s^{(k)}$ can be written as $P_s^{(k)} f(X; W)$ where $P_s^{(k)}$ is also a matrix. We have $\alpha \leq \|I + \lambda L_s\|_2^{-1}$, so $P_s^{(k)}$ will converge to $P_s^*$ as $k$ grows. The output of $\mathcal{D}$ over matrix input is the sum of $\mathcal{D}$ over each row vectors, as typical discriminator function like squared error or cross-entropy do.

Plugging the $Y_s^{(k)}(W)$ and $Y_s^*(W)$ into the loss function, we have

$$\mathcal{L}_{\text{muse}}^{(k)}(W) = \sum_{s=1}^{m} \mathcal{D}\left(Y_s^{(k)}(W)_{1:n_s'}, T_s\right) = \mathcal{D}\left(\mathbb{P}^{(k)} \mathbb{X} W, \mathbb{T}\right) \tag{12}$$

where $1 : n_s'$ means the first $n_s'$ rows, $\mathbb{T} := (T_1, T_2, \dots, T_m)^\top$, $\mathbb{P}^{(k)} := \operatorname{diag}\left\{\left(P_s^{(k)}\right)_{1:n_s'}\right\}_{s=1}^{m}$.

Similarly, for $\ell_W^*(W)$, we change all the $Y_s^{(k)}$ and $P_s^{(k)}$ to $Y_s^*$ and $P_s^*$. That is,

$$\mathbb{P}^* := \operatorname{diag}\left\{(P_s^*)_{1:n_s'}\right\}_{s=1}^{m}, \quad \mathcal{L}_{\text{muse}}^*(W) = \mathcal{D}\left(\mathbb{P}^* \mathbb{X} W, \mathbb{T}\right)$$

**Stochastic Gradient Descent** The updating rule by gradient descent for the parameter is

$$W^{(t+1)} = W^{(t)} - \eta \nabla_{W^{(t)}} \mathcal{L}_{\text{muse}}^{(k)}(W^{(t)}) = W^{(t)} - \eta \sum_{s=1}^{m} \frac{\partial \mathcal{D}\left(Y_s^{(k)}(W^{(t)}), T_s\right)}{\partial W^{(t)}}$$

In reality, we use stochastic gradient descent to minimize $\mathcal{L}_{\text{muse}}^{(k)}(W)$, and the updating rule becomes

$$W^{(t+1)} = W^{(t)} - \eta \frac{\partial \mathcal{D}\left(T_s, g\left(Y_s^{(k)}(W^{(t)})\right)\right)}{\partial W^{(t)}} \tag{13}$$

Here $s$ is picked at random from $\{1, 2, \ldots, m\}$, so the gradient is an unbiased estimator of the true gradient.

**Lemma D.1.** *As defined in* (12), $\mathcal{L}_{muse}^{(k)}(W)$ *is a convex function of* $W$. *Furthermore, there exist the global optimal* $W^{(k*)}$ *such that* $\mathcal{L}_{muse}^{(k)}(W^{(k*)}) \leq \mathcal{L}_{muse}^{(k)}(W)$ *for all* $W$.

*Proof.* $\mathcal{L}_{\text{muse}}^{(k)}(W)$ is a composition of convex function $\mathcal{D}$ with an affine function, so the convexity is retained. By the convexity, the global optimal $W^{(k*)}$ exists. $\square$

**Lemma D.2.** *When the loss function* $\mathcal{L}_{muse}^{(k)}$ *is defined as equation* (12), *and the parameter* $W$ *is updated by* (13) *with diminishing step sizes* $\eta_t = O(\frac{1}{\sqrt{t}})$, $\mathbb{E}\left[\mathcal{L}_{muse}^{(k)}(W^{(t)})\right] - \mathcal{L}_{muse}^{(k)}(W^{(k*)}) = O(1/\sqrt{t})$.

*Proof.* Since the loss function $\mathcal{L}_{\text{muse}}^{(k)}$ is convex by Lemma D.1, and the step size is diminishing in $O(\frac{1}{\sqrt{t}})$, by Nemirovski et al. (2009), the convergence rate of the difference in expected function value and optimal function value is $O(\frac{1}{\sqrt{t}})$. $\square$

**Lemma D.3.** *The subgraph energy function* (6) *with* $\zeta(y) = 0$ *(as defined in Definition 5.1) is* $\sigma_s$-*smooth and* $\tau_s$-*strongly convex with respect to* $Y_s$, *with* $\sigma_s = \sigma_{\max}(I + \lambda L_s)$ *and* $\tau_s = \sigma_{\min}(I + \lambda L_s)$, *where* $\sigma_{\max}$ *and* $\sigma_{\min}$ *are the maximum and minimum singular value of the matrix.*

*Proof.* The proof is simple by computing the Hessian of the energy function. Additionally, since the graph Laplacian matrix $L_s$ is positive-semidefinite, we have $\sigma_s \geq \tau_s > 0$. $\square$

**Lemma D.4.** *Let* $\sigma, \tau = \arg\max_{\sigma_s, \tau_s} \frac{\sigma_s}{\tau_s}, s = 1, 2, \cdots, m$, *where* $\frac{\sigma}{\tau}$ *gives the worst condition number of all subgraph energy functions. In the descent iterations from* (8) *that minimizes* (6) *with step size* $\alpha = \frac{1}{\sigma}$, *we can establish the bound on the propagation matrix* $\|\mathbb{P}^{(k)} - \mathbb{P}^*\| \leq m \exp(-\frac{\tau}{2\sigma}k) \sum_{s=1}^m \|P_s^{(0)} - P_s^*\|$

*Proof.* By Bubeck et al. (2015)[Theorem 3.10], we have

$$\left\| P_s^{(k)} f(X; W) - P_s^* f(X; W) \right\|^2 \leq e^{-\frac{\tau_s}{\sigma_s}k} \left\| P_s^{(0)} f(X; W) - P_s^* f(X; W) \right\|^2$$

Since this bound holds true for any $f(X; W)$, by choosing $f(X; W) = I$ to be the identity matrix, we have

$$\left\| P_s^{(k)} - P_s^* \right\| \leq e^{-\frac{\tau_s}{2\sigma_s}k} \left\| P_s^{(0)} - P_s^* \right\| \leq e^{-\frac{\tau}{2\sigma}k} \left\| P_s^{(0)} - P_s^* \right\|$$

Adding up all the $m$ subgraphs, we have

$$\left\| \mathbb{P}^{(k)} - \mathbb{P}^* \right\| \leq m e^{-\frac{\tau}{2\sigma}k} \sum_{s=1}^m \left\| P_s^{(0)} - P_s^* \right\|$$

$\square$

**Theorem 5.2.** *Let* $W^*$ *be the optimal value of the loss* $\mathcal{L}_{muse}^*(W)$ *per Definition 5.1, while* $W^{(t)}$ *denotes the value of* $W$ *after* $t$ *steps of stochastic gradient descent over* $\mathcal{L}_{muse}^{(k)}(W)$ *with diminishing step sizes* $\eta_t = O(\frac{1}{\sqrt{t}})$. *Then provided we choose* $\alpha \in \left(0, \min_s \|I + \lambda L_s\|_2^{-1}\right]$ *and* $Y_s^{(0)} = f(X_s; W)$, *for some constant* $C$ *we have that*

$$\mathbb{E}\left[\mathcal{L}_{muse}^{(k)}(W^{(t)})\right] - \mathcal{L}_{muse}^*(W^*) \leq O\left(\frac{1}{\sqrt{t}} + e^{-Ck}\right).$$

*Proof.* In Definition 5.1, we assume $\mathcal{D}$ to be Lipschitz continuous for the embedding variable, so for any input $\mathbb{Y}$ and $\mathbb{Y}'$, we have $|\mathcal{D}(\mathbb{Y}, \mathbb{T}) - \mathcal{D}(\mathbb{Y}', \mathbb{T})| \leq L_{\mathcal{D}} \|\mathbb{Y} - \mathbb{Y}'\|$.

$$
\mathcal{L}_{\text{muse}}^{(k)}\left(W^{(k*)}\right) - \mathcal{L}_{\text{muse}}^{*}\left(W^{*}\right)
$$
$$
\leq \mathcal{L}_{\text{muse}}^{(k)}\left(W^{*}\right) - \mathcal{L}_{\text{muse}}^{*}\left(W^{*}\right)
$$
$$
= \mathcal{D}\left(\mathbb{P}^{(k)}\mathbb{X}W^{*}, \mathbb{T}\right) - \mathcal{D}\left(\mathbb{P}^{*}\mathbb{X}W^{*}, \mathbb{T}\right)
$$
$$
\leq L_{\mathcal{D}}\left\|\mathbb{P}^{(k)}\mathbb{X}W^{*} - \mathbb{P}^{*}\mathbb{X}W^{*}\right\|
$$
$$
\leq L_{\mathcal{D}}\left\|\mathbb{P}^{(k)} - \mathbb{P}^{*}\right\|\left\|\mathbb{X}W^{*}\right\|
$$
$$
\leq O\left(e^{-\frac{\tau}{2\sigma}k}\right)
$$

Therefore,

$$
\mathbb{E}\left[\mathcal{L}_{\text{muse}}^{(k)}(W^{(t)})\right] - \mathcal{L}_{\text{muse}}^{*}(W^{*})
$$
$$
= \mathbb{E}\left[\mathcal{L}_{\text{muse}}^{(k)}(W^{(t)})\right] - \mathcal{L}_{\text{muse}}^{(k)}(W^{(k*)}) + \mathcal{L}_{\text{muse}}^{(k)}(W^{(k*)}) - \mathcal{L}_{\text{muse}}^{*}(W^{*})
$$
$$
\leq O(\frac{1}{\sqrt{t}}) + O(e^{-\frac{\tau}{2\sigma}k})
$$
$$
\leq O(\frac{1}{\sqrt{t}} + e^{-ck})
$$

where $c = \frac{\tau}{2\sigma}$. □

### D.3 Alternating Minimization

The main reference for alternating minimization is Csiszár & Tusnády (1984). In this paper, they proved that the alternating minimization method will converge to the optimal value when a five-point property or both a three-point property and a four-point property holds true. We will show that the global energy function and the corresponding updating rule satisfy the three-point property and the four-point property. Thus, the alternating minimization method will converge to the optimal value.

For simplicity, we define $r_{s,i}$ as $r_{s,i} = \sum_{s'=1}^{m}\sum_{j=1}^{n'_s} \mathbb{I}\{I(s,i) = I(s',j)\}$, where $\mathbb{I}\{\cdot\}$ is the indicator function. Therefore, $r_{s,i}$ means the node in the full graph with index $I(s,i)$ appears $r_{s,i}$ times in all subgraphs. With some abuse of the notation, we let $r_{I(s,i)} = r_{s,i}$.

In the energy function (4), we still want to find a set of $\{Y_s\}_{s=1}^{m}$ and $\{\mu_s\}_{s=1}^{m}$ to minimize the global energy function. We can use the alternating minimization method to solve this problem. In each step, we minimize $\{Y_s\}_{s=1}^{m}$ first and then minimize $\{\mu_s\}_{s=1}^{m}$. We want to show that when the parameter $W$ is fixed, by alternatively minimizing $\{Y_s\}_{s=1}^{m}$ and $\{\mu_s\}_{s=1}^{m}$, the energy will converge to the optimal value.

We can easily get the updating rule for $\{Y_s\}_{s=1}^{m}$ and $\{\mu_s\}_{s=1}^{m}$ by taking the derivative of the energy function. Note that we always first update $\{Y_s\}_{s=1}^{m}$ and then update $\{\mu_s\}_{s=1}^{m}$. For $\{Y_s\}$, we have

$$
Y_s^{(k)} = [(1+\gamma)I + \lambda L_s]^{-1}[f(X_s; W) + \gamma \mu_s^{(k-1)}] \tag{14}
$$

For $\{\mu_s\}_{s=1}^{m}$, we have the $i$-th row of $\mu_s^{(k)}$ is

$$
\mu_{s,i}^{(k)} = \frac{1}{r_{s,i}} \sum_{s'=1}^{m} \sum_{j=1}^{n'_s} Y_{s',j}^{(k)} \cdot \mathbb{I}\{I(s,i) = I(s',j)\} \tag{15}
$$

In another word, the updated $\mu^{(k)}$ is the average of the embeddings of the same node in different subgraphs.

Note that here $\{Y_s\}$ with $\{\mu_s\}$ and $\mathbb{Y}$ with $M$ are used simultaneously. We will use $\{Y_s\}$ and $\{\mu_s\}$ when we want to emphasize the subgraphs and use $\mathbb{Y}$ and $M$ when we want to emphasize them as

the input of the global energy function. But in essence they represent the same variables and can be constructed from each other.

The three-point property and four-point property call for a non-negative valued helper function $\delta(Y, Y')$ such that $\delta(Y, Y) = 0$. We define the helper function as $\delta(Y, Y') = (1 + \gamma)\|Y - Y'\|_F^2$. We can easily verify that $\delta(Y, Y) = 0$ and $\delta(Y, Y') \geq 0$.

**Lemma D.5** (Three-point property). *Suppose that we have a series of $\mathbb{Y}^{(k)}$ and $M^{(k)}$, $k = 0, 1, 2, \cdots$ constructed following the updating rule (14) and (15) and that they are initialized arbitrarily. For any $\mathbb{Y}$ and $k$, $\ell_{muse}(\mathbb{Y}, \mu^{(k)}) - \ell_{muse}(\mathbb{Y}^{(k+1)}, \mu^{(k)}) \geq \delta(\mathbb{Y}, \mathbb{Y}^{(k+1)})$ holds true.*

*Proof.* We only need to show $\ell_{muse}^s(Y_s, \mu_s^{(k)}) - \ell_{muse}^s(Y_s^{(k+1)}, \mu_s^{(k)}) \geq \delta(Y_s, Y_s^{(k+1)})$. By iterating $s$ from 1 to $m$ and adding the $m$ inequalities together, we can get the desired result.

$$\ell_{\mathrm{muse}}(Y_s, \mu_s^{(k)}) - \ell_{\mathrm{muse}}(Y_s^{(k+1)}, \mu_s^{(k)}) - \delta(Y_s, Y_s^{(k+1)})$$

$$= \|Y_s - f(X_s; W)\|_F^2 + \lambda \operatorname{tr}\left(Y_s^\top L_s Y_s\right) + \gamma \left\|Y_s - \mu_s^{(k)}\right\|_F^2 - \left\|Y_s^{(k+1)} - f(X_s; W)\right\|_F^2$$

$$\quad - \lambda \operatorname{tr}\left(Y_s^{(k+1)\top} L_s Y_s^{(k+1)}\right) - \gamma \left\|Y_s^{(k+1)} - \mu_s^{(k)}\right\|_F^2 - (1+\gamma) \left\|Y_s - Y_s^{(k+1)}\right\|_F^2$$

$$= \left\|Y_s - Y_s^{(k+1)} + Y_s^{(k+1)} - f(X_s; W)\right\|_F^2 + \lambda \operatorname{tr}\left(Y_s^\top L_s Y_s\right)$$

$$\quad + \gamma \left\|Y_s - Y_s^{(k+1)} - Y_s^{(k+1)} - \mu_s^{(k)}\right\|_F^2 - \left\|Y_s^{(k+1)} - f(X_s; W)\right\|_T^2$$

$$\quad - \lambda \operatorname{tr}\left(Y_s^{(k+1)\top} L_s Y_s^{(k+1)}\right) - \gamma \left\|Y_s^{(k+1)} - \mu_s^{(k)}\right\|_F^2 - (1+\gamma) \left\|Y_s - Y_s^{(k+1)}\right\|_F^2$$

$$= 2 \left\langle Y_s - Y_s^{(k+1)}, Y_s^{(k+1)} - f(X_s; W) \right\rangle + 2\gamma \left\langle Y_s - Y_s^{(k+1)}, Y_s^{(k+1)} - \mu_s^{(k)} \right\rangle$$

$$\quad + \lambda \operatorname{tr}\left(Y_s^\top L_s Y_s\right) - \lambda \operatorname{tr}\left(Y_s^{(k+1)\top} L_s Y_s^{(k+1)}\right)$$

$$= 2 \left\langle Y_s - Y_s^{(k+1)}, (1+\gamma) Y_s^{(k+1)} - \left(f(X_s; W) + \gamma \mu_s^{(k)}\right) \right\rangle$$

$$\quad + \lambda \left\langle Y_s - Y_s^{(k+1)}, L_s \left(Y_s + Y_s^{(k+1)}\right) \right\rangle$$

$$= \left\langle Y_s - Y_s^{(k+1)}, 2 \left(\lambda L_s + (1+\gamma) I\right) Y_s^{(k+1)} - 2 \left(f(X_s; W) + \gamma \mu_s^{(k)}\right) + \lambda L_s \left(Y_s - Y_s^{(k+1)}\right) \right\rangle$$

$$= \left\langle Y_s - Y_s^{(k+1)}, \lambda L_s \left(Y_s - Y_s^{(k+1)}\right) \right\rangle \geqslant 0$$

$\square$

**Lemma D.6** (four-point property). *Suppose that we have a series of $\mathbb{Y}^{(k)}$ and $M^{(k)}$, $k = 0, 1, 2, \cdots$ constructed following the updating rule (14) and (15) and that they are initialized arbitrarily. For any $\mathbb{Y}$, $\mu$ and $k$, $\delta(\mathbb{Y}, \mathbb{Y}^{(k)}) \geq \ell_{muse}(\mathbb{Y}, \mu^{(k)}) - \ell_{muse}(\mathbb{Y}, \mu)$ holds true.*

*Proof.* Expanding all the functions we can get our target is equivalent to

$$\sum_{s=1}^m (1+\gamma) \left\|Y_s - Y_s^{(k)}\right\|^2 \geq \gamma \sum_{s=1}^m \left(\left\|Y_s - \mu_s^{(k)}\right\|_F^2 - \|Y_s - \mu_s\|_F^2\right)$$

We can actually show a stronger result, that is

$$\sum_{s=1}^m \left\|Y_s - Y_s^{(k)}\right\|_F^2 \geq \sum_{s=1}^m \left(\left\|Y_s - \mu_s^{(k)}\right\|_F^2 - \|Y_s - \mu_s\|_F^2\right)$$

By the updating rule for $\mu_s$, we have

$$
\begin{aligned}
&\sum_{s=1}^{m} \left\langle \mu_s - \mu_s^{(k)}, \mu_s^{(k)} - Y_s^{(k)} \right\rangle \\
&= \sum_{s=1}^{m} \sum_{i=1}^{n_s} \left\langle \mu_{s,i} - \mu_{s,i}^{(k)}, \mu_{s,i}^{(k)} - Y_{s,i}^{(k)} \right\rangle \\
&= \sum_{v=1}^{n} \sum_{s=1}^{m} \sum_{i=1}^{n_s} \mathbb{I}\{I(s,i) = v\} \left\langle M_v - M_v^{(k)}, M_v^{(k)} - Y_{s,i}^{(k)} \right\rangle \\
&= \sum_{v=1}^{n} r_v \left\langle M_v - M_v^{(k)}, M_v^{(k)} - \frac{1}{r_v} \sum_{s=1}^{m} \sum_{i=1}^{n_s} \mathbb{I}\{I(s,i) = v\} Y_{s,i}^{(k)} \right\rangle \\
&= 0
\end{aligned}
\tag{16}
$$

Therefore,
$$
\begin{aligned}
&\sum_{s=1}^{m} \left\| Y_s - Y_s^{(k)} \right\|_F^2 \\
&= \sum_{s=1}^{m} \left( \|Y_s - \mu_s\|_F^2 + \left\| \mu_s - Y_s^{(k)} \right\|_F^2 + 2 \left\langle Y_s - \mu_s, \mu_s - Y_s^{(k)} \right\rangle \right) \\
&= \sum_{s=1}^{m} \left( \|Y_s - \mu_s\|_F^2 + \left\| \mu_s - \mu_s^{(k)} \right\|_F^2 + \left\| \mu_s^{(k)} - Y_s^{(k)} \right\|_F^2 \right. \\
&\quad \left. + 2 \left\langle Y_s - \mu_s, \mu_s - Y_s^{(k)} \right\rangle \right) + \sum_{s=1}^{m} 2 \left\langle \mu_s - \mu_s^{(k)}, \mu_s^{(k)} - Y_s^{(k)} \right\rangle \\
&\stackrel{(16)}{=} \sum_{s=1}^{m} \left( 2 \|Y_s - \mu_s\|_F^2 + \left\| Y_s - \mu_s^{(k)} \right\|_F^2 + \left\| \mu_s^{(k)} - Y_s^{(k)} \right\|_F^2 \right) \\
&\quad + 2 \sum_{s=1}^{m} \left( \left\langle Y_s - \mu_s, \mu_s - Y_s^{(k)} \right\rangle + \left\langle \mu_s - Y_s, Y_s - \mu_s^{(k)} \right\rangle \right) \\
&= \sum_{s=1}^{m} \left( 2 \|Y_s - \mu_s\|_F^2 + \left\| Y_s - \mu_s^{(k)} \right\|_F^2 + \left\| \mu_s^{(k)} - Y_s^{(k)} \right\|_F^2 \right) \\
&\quad + \sum_{s=1}^{m} \left( -2 \|Y_s - \mu_s\|_F^2 + 2 \left\langle Y_s - \mu_s, \mu_s^{(k)} - Y_s^{(k)} \right\rangle \right) \\
&= \sum_{s=1}^{m} \left( \left\| Y_s - \mu_s^{(k)} \right\|_F^2 + \left\| \mu_s^{(k)} - Y_s^{(k)} \right\|_F^2 + 2 \left\langle Y_s - \mu_s, \mu_s^{(k)} - Y_s^{(k)} \right\rangle \right) \\
&= \sum_{s=1}^{m} \left( \left\| Y_s - \mu_s^{(k)} \right\|_F^2 + \left\| \mu_s^{(k)} - Y_s^{(k)} + Y_s' - \mu_s \right\|_F^2 - \|Y_s - \mu_s\|_F^2 \right) \\
&\geq \sum_{s=1}^{m} \left( \left\| Y_s - \mu_s^{(k)} \right\|_F^2 - \|Y_s - \mu_s\|_F^2 \right)
\end{aligned}
$$

$\square$

**Theorem 5.3.** *Assume $\zeta(y) = 0$. Suppose we have a series of $\mathbb{Y}^{(k)}$ and $M^{(k)}$, $k = 0, 1, 2, \cdots$ constructed following the updating rule (14) and (15), with $\mathbb{Y}^{(0)}$ and $M^{(0)}$ initialized arbitrarily. Then*

$$
\lim_{k \to \infty} \ell_{muse}(\mathbb{Y}^{(k)}, M^{(k)}) = \inf_{\mathbb{Y}, M} \ell_{muse}(\mathbb{Y}, M),
$$

*Proof.* By Lemma D.5 and Lemma D.6 where both the three-point and four-point property hold true, the theorem is obtained according to Csiszár & Tusnády (1984). $\square$

