# OpenReview forum: "Interpretable and Convergent Graph Neural Network Layers at Scale"
_ICLR.cc/2024/Conference — Submitted to ICLR 2024_

### Official Review · Reviewer_tz3b · 2023-10-30

**Soundness:** 3 good
**Presentation:** 4 excellent
**Contribution:** 3 good
**Rating:** 8
**Confidence:** 3

**Summary:**

The paper describes a novel architecture for graph data. It extends the "unfolded GNN" line of research, where GNNs are formulated by an outer loss function to be minimized, and an inner energy function which is minimized layer-by-layer. In this inner energy function, a "base model" (e.g., an MLP over the nodes) is defined, and the graph layer is defined as a balance between deviating from this base model and smoothing the signal over the graph.

To scale it up, in the proposed MuseGNN the authors combine the idea of sampling subgraphs from the original graph with the unfolded GNN model. A new unfolded model is defined by running the original unfolded model on each subgraph, augmented by an additional regularization term that enforces similarity across subgraphs of the nodes' representations.

They show several convergence analyses on the model of different types. On the experimental side, they show the model is able to achieve better results on very large datasets with slightly higher training time than the baselines.

**Strengths:**

The paper is well written and easy to follow. The idea is described clearly. The model is a combination of two known ideas (unfolded GNNs and subgraph sampling), but it is complemented by a good theoretical analysis and good experimental results. Overall, this is an interesting contribution for the field.

**Weaknesses:**

The biggest weakness of the paper is that the authors keep saying that the model is "interpretable" (e.g., "while maintaining the interpretability attributes of an unfolded GNN"), but this is never truly motivated. The idea is that a user can look at the difference between the base model and the true output to understand whether the prediction was done by looking at the features or at the graph's structure, but this is a very weak notion of "interpretability". In addition, the authors are never showing examples of this.

I have not found any analysis on the memory of the approach, which requires storing predictions and auxiliary embeddings for multiple subgraphs. In addition, the authors should provide an ablation study on the number of subgraphs that are chosen.

In the related works, several papers have proposed subgraph-based GNNs to improve the expressiveness of standard message-passing. Can you compare your approach, either methodologically or experimentally?

**Questions:**

Based on the listed weaknesses:
1. Add a true explainability analyis or remove most of these claims.
2. Add an analysis of the memory required by the model.
3. Add some ablations on the number of chosen subgraphs.
4. Improve the related works section.

---

> ### Author Response · Authors · 2023-11-17
> **Response to Reviewer tz3b**
>
> Thanks for the helpful comments.  We address each point in turn below.
>
> **Comment:**
> *[General Weakness and Question 1.] Add a true explainability analysis or remove most of these claims.*
>
> **Response:**
> Thanks for the suggestion; please see our general response to all reviewers presented above (and we are happy to modify the draft as needed).
>
>
> **Comment:**
> *[General Weakness and Question 2.] Add an analysis of the memory required by the model.*
>
> **Response:**
> Memory requirements were deferred to the supplementary to save space; however, we can summarize here.  For the lightweight $\gamma=0$ case, the memory requirement is the same as  baselines like GCN with (online) neighbor sampling. Namely, $O(nd)$ of memory is required to store the input features in  main memory / disk, and $O(n_sdK)$ of GPU memory is required to store intermediate results (embeddings and gradients) in the device. Here $n$ and $n_s$ are the number of nodes in the full graph and subgraphs, respectively, $d$ is the dimension of input features / hidden embedding, and $K$ is the number of layers. For $\gamma>0$, an extra $O(nd)$ of memory is needed to store the mean vectors $M$, while there is no extra memory needed on the GPU side. As a point of reference, this extra memory requirement is much lower than the $O(ndK)$ required by the GAS-based baselines (Fey et al., 2021) presented in Table 2.
>
>
> **Comment:**
> *[General Weakness and Question 3.] Add some ablations on the number of chosen subgraphs.*
>
> **Response:**
> We note that the number of subgraphs is not a unique issue to MuseGNN, and could potentially affect the performance of any GNN model based on sampling.  Still, this is a useful stability metric to check as the reviewer suggests to rule out any appreciable sensitivity.  As such, we performed such an ablation with ogbn-products data using 50%, 60%, 70%, 80%, 90%, and 100% of the subgraphs that we originally used in the experiments from Table 2.  The accuracy differences were less than 1% across all cases, indicating a degree of robustness to the number of subgraphs.  We can add these results to the supplementary in a later revision.
>
>
> **Comment:**
> *[General Weakness and Question 4.] In the related works, several papers have proposed subgraph-based GNNs to improve the expressiveness of standard message-passing. Can you compare your approach, either methodologically or experimentally?*
>
> **Response:**
> This is an interesting point.  We surmise from our submission that the reviewer may be referring to the reference (Zeng et al., 2021) or related, which discusses subgraph-based sampling that can influence expressiveness by decoupling depth (number of model layers) from scope (the receptive field size).  Similar to Zeng et al., MuseGNN also naturally facilitates the decoupling of depth and scope, where the scope is effectively determined by the underlying sampling method.  With regard to depth though, we differ from Zeng et al. in the sense that additional MuseGNN layers push node embeddings closer and closer to a specific target, namely, a minimizer of the objective from Eq.(4).  Another more subtle distinction exists w.r.t. oversmoothing:  While Zeng et al. relies directly on decoupling to avoid oversmoothing, MuseGNN will be immune to oversmoothing regardless of the particular sampler as long as minimizers of the original energy (4) do not oversmooth.

---

> > ### Comment · Reviewer_tz3b · 2023-11-20
> >
> > Thanks for the answer.
> > 1. On interpretability: it seems 3/4 reviewers (me, ik1J, PP1g) were confused by your use of "interpretability", especially given its focus on the title. I believe the paper can improve significantly be removing this, at least from the title.
> > 2. Subgraph-based NNs: I was referring to the papers by Frasca et. al, 2022 (https://arxiv.org/abs/2206.11140) and following.
> > For the rest, the comments were addressed and I am happy to keep my current evaluation.

---

> ### Author Response · Authors · 2023-11-20
> **Follow-up Response to Reviewer tz3b**
>
> Thanks for the quick response.  Regarding the title, we agree that removing "interpretability" is a good idea.  As a more reflective replacement, one option could be something like: "Forming Scalable, Convergent GNN Layers that Minimize a Sampling-Based Energy."
>
> Also, thanks for pointing to the Frasca et al. (2022) reference. Upon quick inspection, we notice that this paper is rich with details that require further time to digest.  As one speculative possibility though, the ideas from Frasca et al. could potentially be leveraged to more precisely characterize the expressiveness of MuseGNN's sampling-based energy function.

---

### Official Review · Reviewer_ceNB · 2023-10-31

**Soundness:** 2 fair
**Presentation:** 2 fair
**Contribution:** 2 fair
**Rating:** 3
**Confidence:** 3

**Summary:**

The authors of the paper present a new approach to address the scalability issues of Graph Neural Networks (GNNs) when constructed with sampling-based energy functions. The paper also discusses the motivation behind unfolded GNNs, their advantages in terms of interpretability, and the challenges of scaling such models. It introduces the MuseGNN model, which addresses these challenges by incorporating efficient subgraph sampling into the energy function design. It demonstrates increased performance over the baselines on large-scale node classification tasks.

**Strengths:**

1. The paper introduces a novel approach to address the scalability challenges faced by GNNs. The incorporation of offline subgraph sampling into the energy function design is a unique and innovative concept.

2. The paper provides a theoretical analysis of the convergence properties of MuseGNN, when γ is set to 0.

3. The paper includes experimental results showing that MuseGNN performs competitively in terms of accuracy and scalability, especially on large graphs exceeding 1TB in size. This demonstrates the practical feasibility of the proposed approach in large scale graphs.

**Weaknesses:**

1. One notable weak point of the paper's experimental evaluation is its limited comparison to a rather outdated set of baseline models. While the paper does present compelling results in terms of the proposed MuseGNN's performance, the absence of more contemporary and diverse baseline models hinders the thorough assessment of the method's competitiveness and applicability in the current research landscape. Graph neural networks have seen significant advancements in recent years, resulting in a multitude of state-of-the-art models and techniques that offer superior performance across various graph-related tasks. Focusing solely on older and limited baseline models (GCN, GraphSage, GAT) from the past can potentially lead to a skewed perspective of MuseGNN's relative performance in the current state of the art. A more comprehensive comparison against a broader range of modern baseline models would provide a more accurate and up-to-date assessment of the strengths and weaknesses of MuseGNN. I provide some example papers [1,2,3] that can be used as baselines below. Moreover, in the ogb leaderboard https://ogb.stanford.edu/docs/leader_nodeprop/#ogbn-arxiv there are many new baselines.

2. Another weak point in the paper lies in its motivation and justification for employing unfolded GNNs, especially for those who may not be well-versed in this specific research area. While the paper briefly discusses the benefits of using unfolded GNNs and emphasizes their role in enhancing explainability by distinguishing the relative importance of node features and graph structure in predictive tasks, the argument remains somewhat vague and underdeveloped. The paper primarily suggests that the use of unfolded GNNs can help reveal whether node features or graph structure hold more significance for predictions. However, it lacks a more thorough and nuanced discussion of why this is an important aspect to investigate, and how this contributes to the broader field of graph-based machine learning or the potential practical implications.



References:

[1] Rusch, T. Konstantin, et al. "Graph-coupled oscillator networks." International Conference on Machine Learning. PMLR, 2022.

[2]  Rusch, T. Konstantin, et al. "Gradient gating for deep multi-rate learning on graphs. 2022

[3]  Nikolentzos, Giannis, Michail Chatzianastasis, and Michalis Vazirgiannis. "Weisfeiler and Leman go Hyperbolic: Learning Distance Preserving Node Representations." International Conference on Artificial Intelligence and Statistics. PMLR, 2023.

**Questions:**

1. Could the authors provide more context on their choice of limited and older baselines for comparison? Are there more recent or relevant methods that could have been included for a more comprehensive evaluation?

2. The paper claims that using unfolded GNNs enhances explainability, but the argument remains somewhat abstract. Can the authors provide specific instances or use cases where this enhanced explainability has a direct impact on decision-making or model interpretability?

3. Can the authors elaborate on the practical scenarios where distinguishing the importance of node features and graph structure is particularly valuable? How might this knowledge influence real-world applications, and can the paper provide more concrete examples?

4. The paper would benefit from a more comprehensive and well-structured explanation and rationale for the concept of unfolded GNNs.

I am more willing to increase my score, if the authors properly address the above concerns.

---

> ### Author Response · Authors · 2023-11-17
> **Response to Reviewer ceNB (Part I)**
>
> Thanks for the detailed comments.  We address each point in turn below.
>
> **Comment:**
> *[From Weakness 1. and Question 1.] One notable weak point of the paper's experimental evaluation is its limited comparison to a rather outdated set of baseline models ... I provide some example papers [1,2,3] that can be used as baselines below. Moreover, in the ogb leaderboard https://ogb.stanford.edu/docs/leader_nodeprop/#ogbn-arxiv there are many new baselines. Could the authors provide more context on their choice of limited and older baselines for comparison? Are there more recent or relevant methods that could have been included for a more comprehensive evaluation?*
>
> **Response:**
> This is a valid question, and it is definitely true that there now exists a considerable array of different GNN architectures.  The ogbn-arxiv leaderboard and references [1,2,3] are nice examples of this diversity as the reviewer mentions.  However, many/most of these methods have not been integrated with sampling (or historical caching like GAS) or shown to be competitive on truly large graph datasets as is our focus.  Also please kindly note that for the two largest datasets, MAG240M and IGB-full, we are already comparing against the top-ranked homogeneous graph leaderboard model, which is GAT (NS).
>
> As for [1,2,3], these are quite interesting references proposing powerful models (especially for heterophily graphs); however, upon inspection of the empirical evaluations, none of the included benchmarks are especially large or require sampling; in fact, all are amenable to full-graph training such that scalability (our focus) is not a significant factor.  As for top-performing models on OGB leaderboards such as ogbn-arxiv, please see the discussion at the beginning of Section 6.
>
> Of course, we in no way intended to imply that [1,2,3] or top ogbn-arxiv models could not eventually be scaled to our large benchmarks. However, it remains an open research question outside of our scope of how to do so, and would likely require considerable engineering bandwidth and resources that we presently do not have.  Regardless, our central aim is really to scale unfolded GNNs, so in this regard, the metric of success is more tied to the two criteria mentioned at the beginning of Section 6, which our results thus far have achieved.

---

> > ### Author Response · Authors · 2023-11-17
> > **Response to Reviewer ceNB (Part II)**
> >
> > **Comment:**
> > *[Weakness 2. and Question 2.] The paper claims that using unfolded GNNs enhances explainability, but the argument remains somewhat abstract. Can the authors provide specific instances or use cases where this enhanced explainability has a direct impact on decision-making or model interpretability?*
> >
> > **Response:**
> > Please see general comments above to all reviewers which address this issue in-depth and provide two representative cases, e.g., such as scenarios where we would like to differentiate the relative importance of features vs network effects in making predictions as outlined in (Yoo et al., "Less is more: SlimG for accurate, robust, and interpretable graph mining." KDD 2023).  Actually, we intended to cite this work in our original submission but inadvertently missed adding it to Section 2.2.  Overall though, our starting assumption is that prior work has *already* demonstrated the value of GNNs formed from interpretable energy functions (e.g., see other references in Section 2.2); we make no original claims of demonstrating this ourselves.
> >
> >
> > **Comment:**
> > *[Weakness 2. and Question 3.] Can the authors elaborate on the practical scenarios where distinguishing the importance of node features and graph structure is particularly valuable? How might this knowledge influence real-world applications, and can the paper provide more concrete examples?*
> >
> > **Response:**
> > This issue is discussed in more depth in (Yoo et al., "Less is more: SlimG for accurate, robust, and interpretable graph mining." KDD 2023), noting that we will add this reference to Section 2.2 in the revision (we inadvertently forgot it in our hasty original submission as mentioned above).  However, as a quick practical example, in fraud detection scenarios, administrators generally want to know which factors (such as node features or network structure, etc.) might have contributed to a user or transaction being labeled as fraudulent, which can potentially help to instantiate counter-measures or reduce bias.
> >
> >
> > **Comment:**
> > *[Question 4.] The paper would benefit from a more comprehensive and well-structured explanation and rationale for the concept of unfolded GNNs.*
> >
> > **Response:**
> > We agree that the concept of unfolded GNNs, or unfolded deep models more generally, is nuanced and possibly difficult to digest. While we can do our best to improve readability, unfortunately with limited space (we already struggled mightily to fit our draft within the page limit), we have to defer to prior work for comprehensive background.  For reference though, this genre of unfolded modeling (which goes by various naming conventions in the literature) extends  broadly outside the domain of GNNs, covering signal processing, computer vision, and beyond.  As one potentially helpful survey we can add as a citation, please see Liu et al., "Investigating bi-level optimization for learning and vision from a unified perspective: A survey and beyond." IEEE Trans. PAMI 2021.

---

### Official Review · Reviewer_tqgM · 2023-10-31

**Soundness:** 3 good
**Presentation:** 4 excellent
**Contribution:** 3 good
**Rating:** 6
**Confidence:** 3

**Summary:**

Unfold GNNs are those whose forward pass iteratively reduces a graph-regularized energy function of interest. The node embeddings of unfolded GNNs serve as both predictive features and energy function minimizers. This paper proposes a sampling-based energy function and designs a scalable unfolded GNN (MuseGNN). The authors also theoretically analyze the convergence behavior of MuseGNN.

**Strengths:**

1. This paper is well-structured and well-written. I really enjoy reading this paper.
2. The proposed method makes sense to me.
3. The experiment results are good compared to the baselines.

**Weaknesses:**

1. Lack of comparison with representative sampling-based GNNs, such as [a, b].
2. The authors did not conduct experiments on datasets that can illustrate the importance of unfolded GNNs.

***
[a] Chiang, Wei-Lin, et al. "Cluster-gcn: An efficient algorithm for training deep and large graph convolutional networks." Proceedings of the 25th ACM SIGKDD international conference on knowledge discovery & data mining. 2019.

[b] Zeng, Hanqing, et al. "GraphSAINT: Graph Sampling Based Inductive Learning Method." International Conference on Learning Representations. 2019.

**Questions:**

Please see "weakness".

---

> ### Author Response · Authors · 2023-11-17
> **Response to Reviewer tqgM**
>
> Thanks for the constructive comments.  We address each point in turn below.
>
> **Comment:**
> *[Weakness 1.] Lack of comparison with representative sampling-based GNNs, such as Cluster-GCN and GraphSAINT.*
>
> **Response:**
> For the largest-scale benchmarks, neighbor sampling (NS) is typically the method of choice, which is why we adopted it for our baseline comparisons (along with GAS as a more recent popular alternative).  Still, as the reviewer suggests, for completeness it can be beneficial to explore other options.  While we did not have time to run both during the rebuttal period, we provide new results with Cluster-GCN below.
>
> | model       | arxiv      | IGB-tiny   | products   | papers100M | MAG240M    | IGB-full   |
> | ----------- | ---------- | ---------- | ---------- | ---------- | ---------- | ---------- |
> | Cluster-GCN | 68.11±0.7  | 71.03±0.1  | 74.4±0.4   | 54.0±0.6   | OOM        | OOM        |
> | MuseGNN     | 72.50±0.19 | 72.80±0.02 | 80.47±0.16 | 66.82±0.02 | 67.26±0.06 | 60.21±0.18 |
>
> Note that if the reviewer feels it is important, we can easily add GraphSAINT as well to a later revision, although this approach is not seen on the leaderboards for any of the largest datasets.
>
>
> **Comment:**
> *[Weakness 2.] The authors did not conduct experiments on datasets that can illustrate the importance of unfolded GNNs.*
>
> **Response:**
> This is actually true by design; please see our general response to all reviewers presented above which addresses this issue.

---

### Official Review · Reviewer_PP1g · 2023-10-31

**Soundness:** 3 good
**Presentation:** 4 excellent
**Contribution:** 2 fair
**Rating:** 6
**Confidence:** 3

**Summary:**

A sampling-based energy function and scalable GNN layers, MuseGNN, is proposed. Convergence guarantees (under certain assumptions) are provided. Experiments on the large dataset IGB-full and MAG240M demonstrate the scaleability of MuseGNN and its competitive performance with GATs (combined with neighbourhood sampling).

**Strengths:**

- The background of unfolding GNNs, related energy functions, etc. is well explained.
- Advantages of the proposed offline sampling approach are discussed that also enable a convergence analysis.
- Assuming that there exists a unique solution, a theoretical convergence analysis derives a convergence rate of $O(1/\sqrt(t) + \exp(-Ck))$.
- The method enables training on very large graphs and achieves state-of-the-art performance on IGB-full.

**Weaknesses:**

- Novelty: What are the practical benefits of the proposed sampling method over neighbourhood sampling?
- MuseGNN is integrated into the energy function and thus a less general approach than neighborhood sampling, which can be applied to most message passing GNNs.
- In comparison to GAT with neighbourhood sampling, the proposed MuseGNN requires a few more training epochs. Also the performance of GAT with neighbourhood sampling is often competitive.
- Significance intervals are not provided for GAT with neighbourhood sampling on the very large graphs, yet, they are computed for the proposed MuseGNN. The reasoning by the authors are long run times. However, GAT with neighbourhood sampling is reported to be slightly faster than MuseGNN. Thus, significance intervals should also be attainable in this case. Without them, the statement that MuseGNN achieves a new state of the art is not actually accurate.
- The convergence analysis relies on the strong assumption of a unique solution.


Minor points:
- The discussed concept of interpretability seems of minor relevance, as it does not relate to explaining how trained GNNs solve a task. It actually refers more to a concept of consistency over samples. What should the practical benefit of this be?

**Questions:**

-What are the limitations of neighborhood sampling that require the development of the proposed offline sampling scheme? From a practical point of view, what are the actual disadvantages of NS that motivate the development of MuseGNN?
-Please add significance intervals for GAT with neighbourhood sampling for GAT (NS) and SAGE (NS) and GCN (NS).
- What are the number of training epochs for GCN (NS)?
- What are the actual training and inference times for the baselines and MuseGNN? This is also a relevant question because the sampling schemes themselves could take different amounts of time.
- What are the memory requirements of MuseGNN versus the baselines?

---

> ### Author Response · Authors · 2023-11-17
> **Response to Reviewer PP1g (Part I)**
>
> Thanks for the detailed comments.  We address each point in turn below.
>
> **Comment:**
> *[From 1st, 2nd Weakness and 1st Question] Novelty: What are the limitations of neighborhood sampling that require the development of the proposed offline sampling scheme? From a practical point of view, what are the actual disadvantages of NS that motivate the development of MuseGNN?*
>
> **Response:**
> Just to clarify, we are not proposing a new sampling scheme per se.  Rather, we are integrating an existing offline sampling approach within a new role, namely, as the basis for the new energy function from Eq.(4), which then leads to a new scalable unfolded GNN architecture and MuseGNN.  In this sense, the motivation of MuseGNN is unrelated to any disadvantages of NS.  On this last point though, some additional context may be useful.
>
> There are two completely separate aspects of sampling considered in our paper that are relevant to understanding MuseGNN:
> 1. There is the distinction between online vs offline sampling as first mentioned in Section 2.3, where the online version involves dynamically obtaining a new set of random samples *during each training epoch*, while the offline version is predicated on *a fixed set of samples obtained prior to training*.  We chose the latter for MuseGNN, and the reasons are detailed in Section 3.1.
> 2. There is the actual design of the sampler itself, which can be applied in *either* an online or offline fashion.  Neighbor sampling (NS) is arguably the most common choice in the literature, but there are many others such as those used by Cluster-GCN or Shadow-GNN etc.  Regardless, as pointed out in our paper, MuseGNN can adopt *any* such sampler, and the convergence guarantees introduced in Section 5 will still hold.
>
> From this distinction then, we hope the nature of MuseGNN relative to prior uses of sampling is more clear, but we are happy to provide further details if needed.
>
>
> **Comment:**
> *[From 4th Weakness and 1st Question (last part)] Please add significance intervals for GAT with neighbourhood sampling for GAT (NS) and SAGE (NS) and GCN (NS).*
>
> **Response:**
> We apologize that the original submission inadvertently did not provide sufficient details regarding these numbers.  In fact, for the two largest datasets, we relied on numbers from the official IGB/OGB leaderboards for GAT (NS), SAGE (NS), and GCN (NS); we did not compute them all from scratch because of the significant time and cost involved (a common practice in the GNN literature).  For example, just a single training run for these models requires a very large GPU instance (such as a `p4dn.24xlarge` AWS instance) with extra large RAM (over 1TB).  So instead we just focused on computing MuseGNN results for these datasets without bandwidth for multiple runs across all baseline models as would be needed for significance intervals.  We will include these details in the revision.  Thanks for pointing out our oversight.
>
> In contrast, for speed results (i.e., epoch timing) alone we do not require training until convergence. Instead, we only need to average over a few epochs to obtain the results in Table 3, which is much more affordable.
>
>
> **Comment:**
> *[2nd Question] What are the number of training epochs for GCN (NS)?*
>
> **Response:**
> The number of epochs changes from dataset-to-dataset, and can even vary a bit from trial-to-trial due to the use of early stoppage (based on the validation set) as commonly adopted during training. As reference though, for ogbn-arxiv it is roughly 30 epochs, for IGB-tiny about 20 epochs, and for ogbn-papers100M it is about 40 epochs.  Incidentally, MuseGNN is similar (e.g., roughly 40 epochs for ogbn-papers100M).
>
>
> **Comment:**
> *[3rd Question] What are the actual training and inference times for the baselines and MuseGNN? This is also a relevant question because the sampling schemes themselves could take different amounts of time.*
>
> **Response:**
> The average training time for each epoch is reported in Table 3, where MuseGNN is shown to be roughly comparable to the commonly-used NS baselines (the total numbers of epochs are similar; see above).  And indeed, while different sampling schemes may require different amounts of time, NS is arguably the fastest (at least conditioned on those in common use), e.g., in our tests, it is faster than ShadowKHop on baseline GNNs.  Hence there are unlikely to be any significantly faster variants than what we have already shown in Table 3.  Please also see Sections B.2 and B.3 for additional ablations related to sampling.
>
> As for inference, although we did not report specific results in the paper, they follow the same trend as training.  Basically, both MuseGNN and the baseline GNNs all skip the backward pass and remain similar in run-time for the forward inference pass.  However, if the reviewer feels these numbers are important, we can easily add them to a revision.

---

> > ### Author Response · Authors · 2023-11-17
> > **Response to Reviewer PP1g (Part II)**
> >
> > **Comment:**
> > *[3rd Weakness] In comparison to GAT with neighbourhood sampling, the proposed MuseGNN requires a few more training epochs. Also the performance of GAT with neighbourhood sampling is often competitive.*
> >
> > **Response**
> > Admittedly, GAT with neighbor sampling (NS) can achieve competitive accuracy using roughly the same training time per epoch on large datasets, an exception being IGB-full, the very largest dataset. However, GAT (NS) is *not* an unfolded GNN model capable of producing node embeddings that minimize an energy function as is our focus with MuseGNN.
> >
> >
> > **Comment:**
> > *[Last Question] What are the memory requirements of MuseGNN versus the baselines?*
> >
> > **Response:**
> > Memory requirements were deferred to the supplementary to save space; however, we can summarize here.  For the lightweight $\gamma=0$ case, the memory requirement is the same as  baselines like GCN with (online) neighbor sampling. Namely, $O(nd)$ of memory is required to store the input features in  main memory / disk, and $O(n_sdK)$ of GPU memory is required to store intermediate results (embeddings and gradients) in the device. Here $n$ and $n_s$ are the number of nodes in the full graph and subgraphs, respectively, $d$ is the dimension of input features / hidden embedding, and $K$ is the number of layers. For $\gamma>0$, an extra $O(nd)$ of memory is needed to store the mean vectors $M$, while there is no extra memory needed on the GPU side. As a point of reference, this extra memory requirement is much lower than the $O(ndK)$ required by the GAS-based baselines (Fey et al., 2021) presented in Table 2.
> >
> >
> > **Comment:**
> > *[5th Weakness] The convergence analysis relies on the strong assumption of a unique solution.*
> >
> > **Response**
> > Actually, we do not rely on an additional assumption that there exists a unique solution for establishing our convergence results. Regarding Theorem 5.2, given the assumption that the discriminator function $\mathcal{D}$ is convex in Definition 5.1, there could still exist multiple optimal solutions ($W$) with the same loss ($\mathcal{L}^*(W)$). Even so, if there is no unique solution, we still converge to a set of optimal solutions that achieves the minimal possible loss as stipulated by the theorem. And for Theorem 5.3, as the lower-level energy from Eq. (4) always has a unique minimum, no additional assumptions are needed.
> >
> >
> > **Comment:**
> > *[Minor points] The discussed concept of interpretability seems of minor relevance, as it does not relate to explaining how trained GNNs solve a task. It actually refers more to a concept of consistency over samples. What should the practical benefit of this be?*
> >
> > **Response**
> > Please see our general response to all reviewers above where the issue of interpretability is discussed.

---

> > > ### Comment · Reviewer_PP1g · 2023-11-22
> > > **No further questions**
> > >
> > > I thank the authors for their response. I do not have any open questions.
> > > The authors might want to consider revising their paper within the discussion period when they still have the chance, as their strong emphasis on interpretability in the current form seems problematic.
> > >
> > > While I appreciate the contribution by the authors to scale unfolded GNNs to larger graphs, the conceptual novelty with respect to the literature seems limited.
> > > Based on discussions with other reviewers, I will potentially adapt my score at a later stage, as the spread of opinions is quite high.

---

> ### Author Response · Authors · 2023-11-23
> **Follow-up Response to Reviewer PP1g**
>
> Thanks for the follow-up comments. Per the reviewer's suggestion, we have quickly uploaded a new version of the paper with interpretability de-emphasized.  This mainly involved editing Section 2.2 and modifying a few other phrases.  There is now no overlap/conflation with existing GNN work on interpretability, with supporting claims for unfolded GNNs deferred to prior work as needed.  The intended message of our paper is surely much more clear now, good suggestion.
>
> In terms of conceptual novelty, we emphasize that, out of all the possible options for scaling GNNs (and unfolded GNNs in particular), we uniquely offer the following:
> 1. Introduce a novel offline-sampling-based energy formulation, including a penalty term for favoring consistency among the  different node representations with the same node index across sampled subgraphs.  To our knowledge, this form of regularization has never been used before.
> 2. Derive an unbiased estimate of the full graph energy for certain choices of sampling (Proposition 3.1).
> 3. Accommodate new convergence guarantees that are *independent of the specific choice of sampling method itself* (Section 5), and account for sampling and the regularization mentioned above.
> 4. Demonstrate SOTA performance on the two largest publicly-available graph benchmarks, and competitive performance on others using just a single, simple/transparent architecture.

---

### Official Review · Reviewer_ik1J · 2023-11-01

**Soundness:** 3 good
**Presentation:** 3 good
**Contribution:** 2 fair
**Rating:** 3
**Confidence:** 5

**Summary:**

This paper studies how to scale unfold GNNs to large-scale graph benchmarks while maintaining acceptable computational and memory overheads, aligning with common GNN alternatives. Specifically, the authors integrate offline subgraph sampling into the energy function to propose a novel sampling-based energy function and derive convergence guarantees for the novel objective, demonstrating its theoretical feasibility. The authors also empirically demonstrate the effectiveness of MuseGNN constructed using the subgraph sampling-based energy function across datasets of widely varying sizes.

**Strengths:**

1. This article is well-organized.
2. The author has presented sufficient theoretical proof to ensure the convergence of the novel sampling-based energy function during training.
3. The proposed MuseGNN framework achieves state-of-the-art performance on the largest graph benchmark IGB-full.

**Weaknesses:**

1. The paper emphasizes \textbf{Interpretable} in the title and illustrates the explanality of energy minimizers. The key concern is the energy minimization is not aligned with the common concept of GNNs interpretation. For example, some of existing works of interpretable GNNs highlight neighborhood structure leading to the node label classification. The superficial conclusion of node embedding information stemming from the node itself or neighbors makes no sense to the real interpretable applications.
2. In addition, the absence of case study to validate the explanations provided by the energy function raises questions about one of the paper's main claims.
3. The incorporation of the energy minimization into GNNs is not novel. The Dirichilet energy of node embeddings, i.e., the second term in Eq.(2), has been extensively studied in graph domains and GNNs. For example, it has been used to analyze over-smoothing issue in [1]. Following this work, several novel GNNs to optimize this energy function have been proposed. The constraint of closeness to base model embedding, i.e., the first term of Eq. (2), has been implicitly included in models like SIGN [2], SAGN [3], DAGNN[4].

[1] Cai, Chen, and Yusu Wang. "A note on over-smoothing for graph neural networks." arXiv preprint arXiv:2006.13318 (2020).
[2] Rossi, Emanuele, et al. "Sign: Scalable inception graph neural networks." arXiv preprint arXiv:2004.11198 7 (2020): 15.
[3] Sun, Chuxiong, Hongming Gu, and Jie Hu. "Scalable and adaptive graph neural networks with self-label-enhanced training." arXiv preprint arXiv:2104.09376 (2021).
[4] Liu, Meng, Hongyang Gao, and Shuiwang Ji. "Towards deeper graph neural networks." Proceedings of the 26th ACM SIGKDD international conference on knowledge discovery & data mining. 2020.

**Questions:**

1. Please address the previous concerns.
2. Present more results about ablations on $\gamma$, i.e. results on IGB-full, the dataset where traditional GNNs with sampling technique obviously suffer a significant performance degradation, to further demonstrate the effectiveness of $M$.
3. In Proposition 3.1, what is the definition of l(M)? What is the purpose of this proposition?
4. According to my understanding, the main novelty of this paper is to incorporate the shared summary embedding matrix $M$ into energy to facilitate controllable linkage between the multiple embeddings that may exist for a given node appearing in different subgraphs (Because the Formula 6 is a commonly used technique for large-scale training). But existing ablation on $\gamma$ are not sufficient to demonstrate the general effectiveness of this additional constraint term.

---

> ### Author Response · Authors · 2023-11-17
> **Response to Reviewer ik1J (Part I)**
>
> Thanks for the detailed comments.  We address each point in turn below.
>
> **Comment:**
> *[From Weakness 1.] The paper emphasizes Interpretable in the title and illustrates the explainability of energy minimizers. The key concern is the energy minimization is not aligned with the common concept of GNNs interpretation. For example, some of existing works of interpretable GNNs highlight neighborhood structure leading to the node label classification. The superficial conclusion of node embedding information stemming from the node itself or neighbors makes no sense to the real interpretable applications.*
>
> **Response:**
> We apologize for any confusion the title may have caused relative to existing work on interpretable GNNs.  Please see our general comments to all reviewers above which directly address this issue.  In any event, we are happy to modify the title and framing to increase clarity.
>
>
> **Comment:**
> *[From Weakness 2.] In addition, the absence of case study to validate the explanations provided by the energy function raises questions about one of the paper's main claims.*
>
> **Response:**
> Please see the general comments above to all reviewers which address this issue in depth.  But in brief, our starting assumption is that prior work has *already* demonstrated the value of unfolded GNNs formed from interpretable energy functions; we make no claims of demonstrating this ourselves.  Instead, our focus herein is entirely on scaling this class of GNN models to the largest possible graphs, with provable convergence guarantees.
>
>
> **Comment:**
> *[From Weakness 3.] The incorporation of the energy minimization into GNNs is not novel.*
>
> **Response:**
> Absolutely, we 100% agree. These models were originally invented, motivated, and validated in prior work, e.g., the many references we cite in Sections 2.1 and 2.2.  That being said, our contribution is *not* the invention or motivation of these models, rather, it is scaling these models to huge graphs with convergence guarantees that account for the required sampling involved.
>
>
> **Comment:**
> *[From Weakness 3.] The Dirichilet energy of node embeddings, i.e., the second term in Eq.(2), has been extensively studied in graph domains and GNNs. For example, it has been used to analyze over-smoothing issue in [1]. Following this work, several novel GNNs to optimize this energy function have been proposed.
> \[1\] Cai, Chen, and Yusu Wang. "A note on over-smoothing for graph neural networks." arXiv preprint arXiv:2006.13318 (2020).*
>
> **Response:**
> Exactly, the Dirichlet energy, often in combination with other terms as in Eq.(2) of our submission, is the starting point for inspiring a wide variety of existing unfolded GNN architectures, e.g., see references in Section 2.1.  And indeed as the reviewer correctly suggests (and the reviewer provided reference [1] shows), this Dirichlet energy can be used to analyze oversmoothing.  In fact, this is a good example of the very type of interpretability that these energy-based models possess.  In this case, the Dirichlet energy alone (akin to the second term in Eq.(2)) leads to oversmoothing since constant node embeddings (i.e., oversmoothing) will achieve the minimum.  In contrast, it has also been observed in prior work that when additional terms involving a base model are included as in Eq.(2), oversmoothing no longer happens precisely because global minimizers transparently no longer oversmooth. Thanks for pointing out reference [1], this is a great example to add to our related work.
>
>
> **Comment:**
> *[From Weakness 3.] The constraint of closeness to base model embedding, i.e., the first term of Eq. (2), has been implicitly included in models like SIGN [2], SAGN [3], DAGNN[4].*
>
> **Response:**
> While these architectures may incorporate some form of base model or precomputed features, to our knowledge the forward pass in each case does not actually minimize an explicit energy function.  Hence we cannot directly interpret their output embeddings as energy function minimizers as we can with the GNN architectures formed by minimizing Eq.(2).  Even so, they can be cited for added context if the reviewer suggests.

---

> ### Author Response · Authors · 2023-11-17
> **Response to Reviewer ik1J (Part II)**
>
> **Comment:**
> *[From Question 2.] Present more results about ablations on $\gamma$, i.e. results on IGB-full, to further demonstrate the effectiveness of $M$.*
>
> **Response:**
> IGB-full is massive and we do not have the time or computing resources to conduct a careful ablation during the short rebuttal period.  However, for now, we can nonetheless use IGB-tiny and IGB-medium as surrogates, as well as ogbn-products as another reference point (note that IGB-medium is actually quite large, with more labeled nodes than even ogbn-papers100M).  From the new ablation table below, we observe that it is indeed possible to improve performance with $\gamma > 0$ in all three cases. (Note that for ogbn-products, this accuracy is actually better than reported in our original submission, where we did not carefully explore tuning $\gamma$.)
>
> | dataset       | $\gamma=0$ | $\gamma=0.1$ | $\gamma=0.5$ |
> | ------------- | ---------- | ------------ | ------------ |
> | IGB-tiny      | 72.66      | 72.78        | **72.81**    |
> | IGB-medium    | 75.18      | 75.80        | **75.83**    |
> | ogbn-products | 80.42      | 80.92        | **81.23**     |
>
>
> **Comment:**
> *[From Question 3.] In Proposition 3.1, what is the definition of $\ell(M)$? What is the purpose of this proposition?*
>
> **Response:**
> The function $\ell$ is defined by Eq.(2), and $\ell(M)$ merely refers to Eq.(2) with $M$ as the input argument.  The paragraph directly above Proposition 3.1, where we describe how the role of $M$ equates to $Y$ when $\gamma = \infty$, provides context for why we introduce $\ell(M)$.
>
> In terms of purpose, Proposition 3.1 serves a key role by quantifying the relationship between ideal full-graph training as instantiated via Eq.(2), and offline sampling using Eq.(4) as used by MuseGNN in our submission.  More precisely, when $\gamma = \infty$, Eq.(4) reduces to Eq.(7).  From here, Proposition 3.1 then proves that Eq.(7) becomes equivalent to Eq.(2) in expectation (i.e., it is unbiased), provided we conduct sampling independently as described in the proposition statement. We hope this helps to clarify the value of Proposition 3.1, and we are happy to update the draft if necessary to do so. (The end of Section 3.2 very briefly mentions these ideas, but further detail could be helpful.)
>
>
> **Comment:**
> *[From Question 4.] According to my understanding, the main novelty of this paper is to incorporate the shared summary embedding matrix $M$ into energy to facilitate controllable linkage between the multiple embeddings that may exist for a given node appearing in different subgraphs (Because the Formula 6 is a commonly used technique for large-scale training).*
>
> **Response:**
> Our novelty extends beyond incorporating the shared summary embedding matrix $M$ into the stated energy as follows:
> * Although offline sampling has been proposed previously, to our knowledge Eq.(6) has not been used in prior work as an energy function for producing GNN layers through minimization (our focus). Please also see Section 3.1 for further details regarding why we chose offline sampling, which differs from prior work.
> * Our convergence results, which cover both the $\gamma = 0$ case, meaning Eq.(6), and the more general $\gamma > 0$ case are new.  Moreover, by relying on offline sampling these results are agnostic to the underlying sampling method itself.
> * Finally, even if we exclude consideration of the empirical efficacy of $\gamma > 0$, Proposition 3.1 quantifies a precise regime whereby our sampling-based energy provides an unbiased estimator of the full-graph energy.  This is notable because, generally speaking, it is infeasible to establish notions of unbiasedness in arbitrary GNN models instantiated with sampling (with the exception of unbiasedness narrowly defined within a single model layer).
>
>
> **Comment:**
> *[From Question 4.] But existing ablation on $\gamma$ are not sufficient to demonstrate the general effectiveness of this additional constraint term.*
>
> **Response:**
> Regarding additional ablations, please see our response to Question 2 above.

---

### Author Response · Authors · 2023-11-17
**General Response to All Reviewers**

Thanks to all reviewers for the constructive comments.  To minimize redundancy, we provide some general feedback here that is relevant to multiple reviews.

**Different Notions of Interpretability.** There are many different notions of what reasonably constitutes GNN interpretability or explainability.  Indeed the GNN literature includes a variety of modeling frameworks that purport to *explain* which factors are likely to influence predictions, e.g., GNNexplainer, GNN-LRP, XGNN.  However, within the context of this submission, our intended notion of interpretability is quite specific, and admittedly somewhat different from prior notions: we are addressing GNN architectures designed with layers that are in one-to-one correspondence with descent iterations that minimize a transparent, *interpretable* energy function.  Or stated differently, as the node embeddings of this class of GNN architecture are updated from layer-to-layer during the forward pass, they can be viewed as descending the aforementioned interpretable energy function.  In this way, potentially-useful properties of these node embeddings can be inferred from the intrinsic characteristics of the underlying energy function.

For example, if a minimizer of the energy is robust against spurious edges in a graph, then by design a GNN architecture constructed in this way will likely be robust against corrupted graphs (Yang et al., 2021).  Other designs may target differentiating the relative importance of features vs network effects in making predictions; see (Yoo et al., "Less is more: SlimG for accurate, robust, and interpretable graph mining." KDD 2023) for motivation. (Note that we inadvertently forgot to add this reference to Section 2.2 of our submission, but will do so upon revision.)

**Choice of Title/Framing.** Because of the aforementioned energy function association, we chose to include the phrase "interpretable and convergent" in our paper title as shorthand, with the goal of describing our more nuanced meaning in the abstract and introduction.  However, now with the benefit of hindsight (and valuable reviewer feedback), perhaps this was ultimately not the best choice.  Given that interpretability already has a strong prior usage in the GNN community, our title phrasing may introduce unnecessary conflation with past work and we are willing to change it if reviewers prefer.  And critically, empirical demonstration of interpretability is not even the point of our paper, so we want to avoid unnecessary diversions.

**Clarifying Paper Scope.** Quite differently, the utility of GNN architectures formed by unfolded optimization steps as we describe above (and in-depth in the paper), is *already* well-established with a vast supporting literature, including examples of how transparent properties of the underlying energy function can serve practical needs, e.g., the references we list in Section 2.2 and others we can add (more are popping up all the time).  In this sense, we intentionally and exclusively rely on *prior work* to support these claims and motivate what is actually our primary contribution below.

**Concise Statement of Primary Contribution.** Conditioned on the value of this class of unfolded GNN architecture (as supported by extensive prior references, *not* our own work), our primary contribution is to demonstrate for the first time:

* How to scale these models to the largest publicly-available graph benchmarks using offline sampling embedded within the energy function design itself (Section 3.2), while still retaining the desirable properties that made this class of architectures attractive in the first place.
* Derive explicit connections between our proposed framework and ideal full-graph or sampling-free training (Section 3.3);
* For our proposed sampling-based architecture MuseGNN and training procedure (Section 4), establish convergence guarantees (Section 5) that form a sturdier foundation and help raise our confidence in reliable performance were these models to be deployed in novel domains;
* Showcase competitive node classification performance on the largest possible publicly available datasets (Section 6); and for the two largest, MAG240M and IGB-full we reach SOTA among GNNs as very few systems can handle this scale.

We hope that these remarks help to clarify our position, and we apologize for any unintended ambiguity that our original title and framing may have caused.

---

> ### Author Response · Authors · 2023-11-23
> **A Note on the Revised Paper**
>
> To reflect the de-emphasis of "interpretability", we have now uploaded a new draft of the paper per Reviewer PP1g's request (any other reviewer suggested changes can easily be incorporated later, as time is now limited and the discussion window will soon close). The required updates mainly involved editing Section 2.2 and modifying a few other phrases.  There is now no overlap/conflation with existing GNN work on interpretability, with supporting claims for unfolded GNNs deferred to prior work as needed.  Thanks to reviewers for constructively pointing out this source of confusion, and the current openreview draft is notably improved because of it.

---

### Meta-Review · Area_Chair_jTb3 · 2023-12-19

**Metareview:**

As with its original title, this paper aims to build an interpretable GNN through unfolded GNNs and enhance its scalability through sampling. Unfolded GNNs are a class of GNN methods derived from an interpretable optimization objective (aka energy function), which may face scalability issues with too many iteration steps. This paper aims to resolve this problem by sampling techniques. The authors also put theoretical guarantees on the convergence properties of subsampled unfolded GNNs. Experiments on multiple large scale datasets show that the proposed method attain good performance on par with the full-batch version, and indeed resolve the scaling problems of full-batch GNNs.

This paper receives a quite diverse set of reviews. On the positive side, reviewers generally appreciate the theoretical analysis and empirical performance of the proposed approach. However, on the negative side, there are two reviewers (ik1J, ceNB) that held strong opinions towards rejections. Both of them critiqued the motivation and justification for adopting unfolded GNNs. The claimed GNN interpretation may conflict with existing literature, while the interpretability that the authors claim, also has limited practical significance. Meanwhile, given the existing litearture, concerns on limited novelty of the proposed unrolled GNN objectives compared to previous ones. Reviewer CeNB also raises concerns on outdated baselines  have also been raised. During the rebuttal, the authors modify the paper title to emphasize scalability instead of interpretability, which is good, but seems still not enough to ease the reviewers’ concerns.

Summarizing these diverse opinions, I do think that the current presentation of the paper lacks a clear justification of the proposed approach, and the new objective also has limited novelty compared to previous ones. These limitations hinder this work to meet the ICLR bar. Nevertheless, since the paper does deliver quite impressive empirical performance, the authors are encouraged to revise and resubmit it to future venues.

Lastly, as a side note, I would also like to point out a few papers that also proposed sampling techniques for scaling unfolded/convergent GNNs [1,2] before this work. Although both the reviewers and the authors do not bring them up, the sampling techniques are indeed highly related and worth discussing in future revisions.

[1] Li et al. Unbiased Stochastic Proximal Solver for Graph Neural Networks with Equilibrium States. ICLR 2023.

[2] Chen et al. Efficient and Scalable Implicit Graph Neural Networks with Virtual Equilibrium. IEEE BigData 2022.

**Justification For Why Not Higher Score:**

The proposed work has limited technical novelty and also lacks a clear motivation of the proposed approach, which could be refined in future updates.

**Justification For Why Not Lower Score:**

N/A

---

### Decision · Program_Chairs · 2024-01-16

Reject